# Skill Discovery using Language Models

## Abstract

Large Language models (LLMs) possess remarkable ability to understand natural language descriptions of complex robotics environments. Earlier studies have shown that LLM agents can use a predefined set of skills for robot planning in long-horizon tasks. However, the requirement for prior knowledge of the skill set required for a given task constrains its applicability and flexibility. We present a novel approach L2S (short of Language2Skills) to leverage the generalization capabilities of LLMs to decompose the natural language task description of a complex task to definitions of reusable skills. Each skill is defined by an LLM-generated dense reward function and a termination condition, which in turn lead to effective skill policy training and chaining for task execution. To address the uncertainty surrounding the parameters used by the LLM agent in the generated reward and termination functions, L2S trains parameter-conditioned skill policies that performs well across a broad spectrum of parameter values. As the impact of these parameters for one skill on the overall task becomes apparent only when its following skills are trained, L2S selects the most suitable parameter value during the training of the subsequent skills to effectively mitigate the risk associated with incorrect parameter choices. During training, L2S autonomously accumulates a skill library from continuously presented tasks and their descriptions, leveraging guidance from the LLM agent to effectively apply this skill library in tackling novel tasks. Our experimental results show that L2S is capable of generating reusable skills to solve a wide range of robot manipulation tasks.

## 1 Introduction

In recent years, the integration of language models with robotics has opened up new avenues for advancing autonomous learning in robotic systems. Large Language models (LLMs) possess the remarkable ability to understand complex tasks and environments. Leveraging this capability, researchers have explored the use of language models in various aspects of robotics, ranging from task planning and navigation to manipulation and control. Previous work in this domain has primarily focused on leveraging language models for robot planning, where a predefined set of skills is provided to the model. However, this approach has limitations, as it assumes prior knowledge of the skill set required for the given task, thus constraining its applicability and flexibility.

Automatic skill acquisition has long been studied in the context of hierarchical reinforcement learning (Barto and Mahadevan (2003)) in the form of temporally extended actions Sutton et al. (1999). Despite the proven effectiveness of skills in expediting learning (McGovern and Sutton (1998)), a fundamental question remains: how can agents autonomously develop valuable skills through interaction with their environment? There has been a significant body of work aimed at discovering skills. For example, Option-Critic (Bacon et al. (2017)) learns skills by optimizing the skill policies as well as their termination functions in a gradient-based manner, assuming all the skills can be applied everywhere. However, it is known to be prone to inefficient task decomposition, such as learning a sub-policy that terminates at every time step or discovering one efficient sub-policy that executes throughout the entire episode. Vezhnevets et al. (2017); Nachum et al. (2018); Levy et al. (2019) address this issue by automatically decomposing a complex task into subtasks and solving them by optimizing the subtask objectives. These methods excel in learning multiple levels of policies in sparse reward tasks. However, the low-level skills learned are tied to a specific environment and it is unclear whether they are adaptable to new tasks. Skill chaining (DSC) (Konidaris and Barto (2009); Bagaria and Konidaris (2020)) involves a sequential discovery and chaining of skills, starting from

the end goal state and progressing backward to the initial state. However, as the agent generates new skills using the initial states of the preceding skill on the skill chain as their goal states, this poses challenges in robot manipulation tasks. For example, learning a skill $\pi_1$ to move an object towards a goal region cannot be learned well before mastering the skill $\pi_2$ for object grasping, but skill chaining would require learning $\pi_1$ first.

We present a novel approach L2S (short for "Language to Skills") for skill discovery in robot learning by leveraging large language models to overcome the limitations of prior methods. We aim to empower robotic systems to autonomously discover and adapt skills to a wide range of tasks. L2S harnesses the generalization capability of large language models (LLMs) to decompose the natural language task description of a complex task to definitions of reusable skills. Each skill is defined by an LLM-generated *dense* reward function and a termination condition, which in turn lead to effective skill policy training and chaining for task execution. For example, consider the "turn faucet left" task depicted in Fig. 1. The GPT-4 agent can break down this task into two skills: (1) positioning the robot's end effector near the right side of the faucet $\pi_{o_1}$ and (2) rotating the faucet handle to the left $\pi_{o_2}$. Chaining these two skills together successfully solves the task.

**L2S excels in sequential task learning** by autonomously building a library of parameterized skills (explained below) as it encounters tasks during training. This accumulated skill library can then be reused to tackle new tasks, guided by the LLM agent. For example, consider a scenario where the agent is presented with the task of "turn faucet right" after it has already been trained on "turn faucet left". The LLM agent identifies that the first skill $\pi_{o_1}$ in "turn faucet left" can be tuned to position the end effector on the left side of the faucet handle (by adjusting its parameters). Thus, L2S only needs to train a new skill $\pi'_{o_2}$ to rotate the faucet handle right. By reusing existing skills in this manner, L2S significantly reduces the computational burden associated with learning new tasks from scratch, enabling more efficient task solving over time.

**The main challenge** faced by L2S is that while LLMs can outline the overall structure of skills necessary to tackle a task, they lack detailed insight into the specific low-level control intricacies of the environment. For the "turn faucet left" task in Fig. 1, the reward function generated for the first skill $\pi_{o_1}$ encourages the skill policy to guide the end effector towards the right side of the handle by a distance of `params[0] = 0.01`m. However, training the policy using this reward function could lead to an unforeseen outcome where the end effector ends up on the left side of the handle, rendering the subsequent skill of rotating the faucet handle left unattainable (Fig. 1 top right). This discrepancy arises from the norm function employed in the reward and termination functions of $\pi_{o_1}$, which solely emphasizes the proximity of the end effector to the `target_position` that is located too close to the faucet handle (at a distance of `params[0] = 0.01`m). Consequently, the policy may position the end effector on the left side of the handle and still achieves a high task reward and satisfies the termination condition of this skill. To address the uncertainty surrounding the parameters used by the LLM agent, L2S trains parameter-conditioned skill policies, denoted as $\pi_o(a|s; \text{params})$, where the parameters `params` are akin to "goals" in goal-conditioned reinforcement learning. As the impact of these parameters on the overall task becomes apparent only when subsequent skills are trained, L2S adopts a strategy of training a skill policy that performs well across a broad spectrum of parameter values and selects the most suitable parameter value during the training of subsequent skills. For instance, the first skill $\pi_{o_1}(a|s; \text{params})$ for "turn faucet left" is trained to position the end effector around the faucet handle, with a distance to the handle at `params[0]`. Training the subsequent skill $\pi_{o_2}$ involves determining the correct policy parameter `params[0]` - the `target_position` to move the end effector to - and appropriately setting its termination condition parameters `t_params[0]` - determining how close the end effector should be to the target position before transitioning to the next skill - to optimally achieve the highest reward during the training of the second skill (Fig. 1 middle). In this way, L2S effectively mitigates the risk associated with potentially incorrect parameter choices. The parameter-conditioned skill policies in L2S facilitate seamless skill reuse. The parameter `params[0]` in the first skill $\pi_{o_1}(a|s; \text{params})$ trained for "turn faucet left" can be adjusted to position the robot's end effector on the left side of the faucet for the "turn faucet right" task.

Compared to state-of-the-art LLM-guided reward generation methods such as Text2Reward Xie et al. (2023) and Eureka Ma et al. (2023), which generate dense reward functions to train single, monolithic policies for each robotic task, L2S instead creates reusable, parameterized skills for *sequential task learning*. These skills effectively generalize to new tasks through parameterization. While previous work, such as Ahn et al. (2022), has explored decomposing complex tasks into

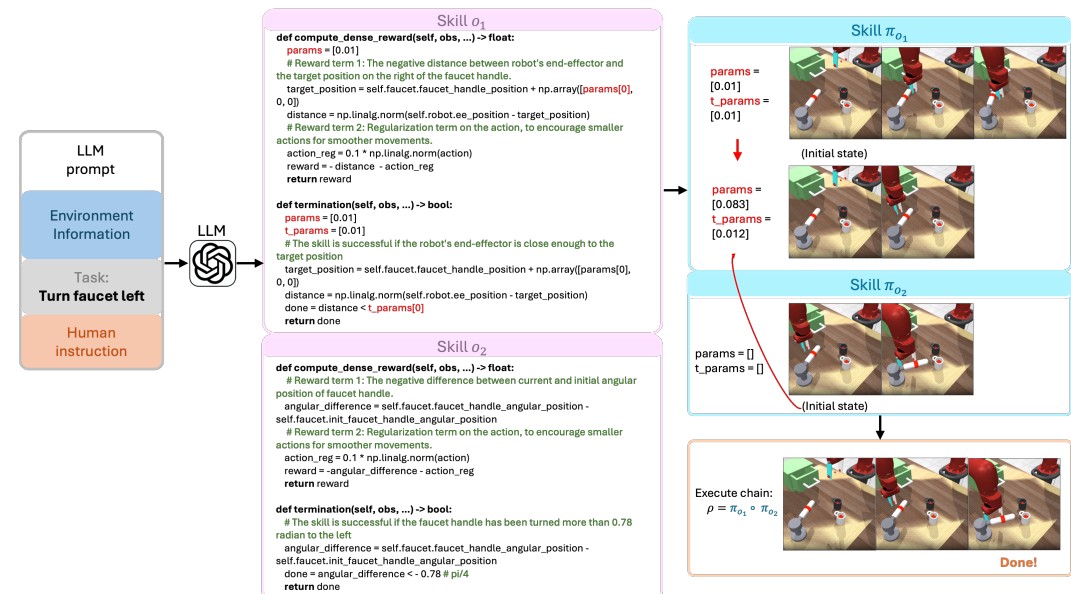

Figure 1: An example "Turn faucet left" to explain the workflow of L2S. The training of skill $\pi_{o_2}$ optimizes the policy and termination parameter of skill $\pi_{o_1}$.

skills using LLMs' semantic knowledge, it relies on manually engineered skill libraries, whereas L2S autonomously learns such a library with parameterization to enable efficient generalization. Our experimental evaluations on a suite of robotics manipulation tasks show that L2S not only solves continuously presented tasks much faster but also achieves higher success rates compared to state-of-the-art methods.

## 2 PROBLEM DEFINITION

Sequential decision-making problems can be formalized as Markov Decision Processes (MDPs). An MDP is defined by a tuple $e = \langle S, A, T, R, \gamma, \eta \rangle$, where $S$ represents the state space, $A$ represents the action space, $T : S \times A \times S \to [0, 1]$ denotes the transition function, $R : S \times A \to \mathbb{R}$ denotes the reward function, $0 < \gamma < 1$ is the discount factor, and $s_0 \sim \eta(\cdot)$ defines the initial states. At each time step $t$, the agent selects an action $a_t \in A$ in state $s_t \in S$, receives a reward $r_t = R(s_t, a_t) \in \mathbb{R}$, and transitions to another state $s_{t+1}$ with a probability determined by $T$. We assume **sparse reward** functions that provide signals only upon task success (1.0) or failure (0.0). The primary objective is to learn a policy $\pi : S \to A$ for $e$ that maximizes the expected return, defined as the discounted sum of rewards: $\max_{\pi \in \Pi} \mathbb{E}_\pi \left[ \sum_{t=0}^\infty \gamma^t R(s_t, a_t) \right]$, where $a_t = \pi(s_t)$.

**Skills**. A significant challenge for reinforcement learning (RL) algorithms lies in learning and planning over long horizons, particularly in scenarios where rewards are sparse. The options framework, proposed by Sutton et al. (1999), offers a formalism for temporal abstraction, which aids in both exploration and credit assignment. The central concept is to decompose the overarching problem that the agent seeks to solve into subtasks, each typically characterized by its own reward function and capable of being accomplished by a distinct skill. Our method L2S is inspired by the options framework and we define skills similar to options in the options framework. A skill $o$ consists of (a) its termination condition, $\beta_o(s)$, which determines whether skill execution must terminate in state $s$ and (b) its closed-loop skill policy, $\pi_o(s)$, which maps state $s$ to a low level action $a \in A$.

**Skill Chaining for Single-Task Learning**. Given a *single* task MDP $e$, and its task description $\mathcal{L}_e$ in natural language, L2S constructs a chain of skills Konidaris and Barto (2009); Bagaria and Konidaris (2020) such that successful execution of each skill in the chain allows the agent to execute another skill. A task description $\mathcal{L}_e$ refers to a language command describing the desired goal for the agent, like "turn faucet left". The inductive bias of creating sequentially executable skills guarantees that as long as the agent successfully executes each skill in its chain, it can solve the original task in $e$. Intuitively, skill chaining amounts to learning skills such that the termination

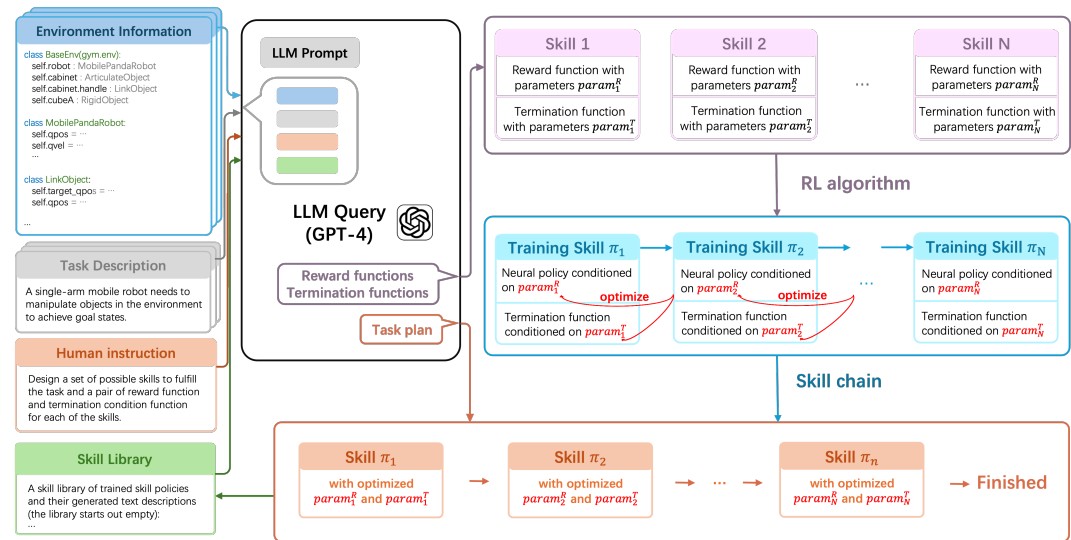

Figure 2: The L2S overall framework.

condition $\beta_{o_i}$ of a skill $o_i$ induces an initiation condition of the skill that follows it in the skill chain. We formally define skill chaining as follows. A skill chain $\rho = o_0 \circ o_1 \circ \ldots \circ o_k$ defines a *controller* $\pi_\rho = (\pi_{o_0}, \beta_{o_0}) \circ (\pi_{o_1}, \beta_{o_1}) \circ \ldots \circ (\pi_{o_k}, \beta_{o_k})$ that navigates from an environment initial state of an MDP $e$ to a state where $\beta_{o_k}$ (the termination condition of $o_k$) holds. In particular, $\pi_\rho$ executes $\pi_{o_j}$ (starting from $j = 0$) until reaching $\beta_{o_j}$, after which it increments $j \leftarrow j + 1$ (unless $j = k$). Note that $\pi_\rho$ is stateful since it internally keeps track of the index $j$ of the current skill policy.

**Skill Library Construction in Sequential Task Learning**. The main objective of L2S is to efficiently tackle a sequence of related tasks by autonomously building a skill library from ongoing tasks and reusing it for future tasks. Formally, given a *sequence* of tasks, where each task is represented as an MDP $e$ and accompanied by a description $\mathcal{L}_e$, L2S builds a skill library $\mathcal{O} \equiv \{o, L_o\}$ tailored for solving these tasks. Each skill $o$ within $\mathcal{O}$ is associated with a descriptive text $L_o$. ***The skill library*** $\mathcal{O}$ ***starts out empty***. As L2S encounters new tasks $(e, \mathcal{L}_e)$ in the sequence, it progressively adds new skills in the skill chains for solving these tasks to $\mathcal{O}$, while also developing plans that make use of the the existing skills in $\mathcal{O}$ whenever possible.

## 3 LANGUAGE TO SKILLS

**The primary goal** of L2S is to utilize LLMs to automatically build skill libraries $\mathcal{O}$ for sequential task learning. Given the textual description of a new task, L2S uses LLMs to generate code that defines both the reward function and termination condition for each new skill in the skill chain for solving the task, facilitating the learning of the skill's policy. These learned skills, along with their LLM-generated descriptions, are subsequently added to $\mathcal{O}$, enabling the LLMs to effectively reuse them when building skill chains for future tasks.

### 3.1 PROMPT CONSTRUCTION

For a task MDP $e$ and its natural language task description $\mathcal{L}_e$, L2S prompts an LLM agent with an abstraction of $e$ and $\mathcal{L}_e$ to generate a skill chain for solving the task. The environment abstraction is needed by the LLM agent to ground reward generation for understanding how object states are represented, including robot and object configurations. We adopt a compact Pythonic representation, similar to Xie et al. (2023), as illustrated in Fig. 2. This approach offers a higher level of abstraction compared to listing all environment-specific information in a table or list format. The LLM agent is instructed to generate a skill chain for $e$ as $\rho = o_0 \circ o_1 \circ \ldots \circ o_k$, and the reward function $\mathcal{R}_{o_i}[\phi_i](s, a)$ and termination condition $\beta_{o_i}[\varphi_i](s)$ (as python programs) for each skill $o_i$ in $\rho$, where $\phi_i$ and $\varphi_i$ are the parameters within the skill reward and termination functions for $o_i$ respectively. Additionally, L2S asks the LLM agent to generate a description $\mathcal{L}_o$ for each skill $o$.

For continuously presented tasks, as discussed in Sec. 2, L2S maintains a skill library $\mathcal{O} \equiv \{(o, \mathcal{L}_o)\}$ where each skill $o$ is accompanied with its text description $\mathcal{L}_o$ (generated by the LLM agent). L2S starts with no predefined skills, meaning **the skill library $\mathcal{O}$ is initially empty**. Skills within the skill chain devised for one task are added to $\mathcal{O}$ for reusing them when building skill chains for future tasks. Thus, it is possible part of a generated skill chain $\rho = o_0 \circ o_1 \circ \ldots$ reuses skills in $\mathcal{O}$. To achieve this, L2S prompts the LLM agent with the text description $\mathcal{L}_o$ of each skill $o$ in $\mathcal{O}$, along with a command instructing the LLM agent to select and reuse existing skills whenever possible.

Although LLMs have good understanding of high-level task structures, we have found that they are not yet reliable enough to generate correct rewards and termination conditions in a zero-shot manner for complex tasks. To handle this, we also prompt LLMs with few-shot examples as Xie et al. (2023). Detailed prompt examples can be found in the Appendix G.

## 3.2 SKILL CHAIN TRAINING

During training, L2S iteratively processes(learning and optimizing, if necessary) skills in a skill chain $\rho$ starting from the initial skill $o_0$, continuing until it processes the final skill on $\rho$. It maintains the property that for every skill $o_i$ it processes, it has already trained policies for all skills preceding $o_i$.

The key challenge with this approach is that, although LLMs can define the overarching structure of a skill chain for a given task, they often lack precise knowledge of the low-level control details within the environment. As a result, the parameters used in the generated reward and termination functions tend to be inaccurate, reflecting inherent uncertainties (as illustrated in the turn faucet example in Sec. 1 and Fig. 1). An important aspect of L2S is that each skill policy is parameter-conditioned (similar in concept to goal-conditioned reinforcement learning), denoted as $\pi_{o_i}(a|s; \phi_i)$, where $\phi_i$ represents the parameters in the reward function $\mathcal{R}_{o_i}$ for $o_i$. The training objective is for the policy $\pi_{o_i}(a|s;\phi_i)$ to maximize the expected rewards over a broad range of parameter values $\phi_i \sim \tilde{q}_{\phi_i}$, ensuring robust performance across varying conditions. The parameter distribution $\tilde{q}_{\phi_i}$ for $\phi_i$ is configured by the user. For example, one can set $\tilde{q}_{\phi_i}$ as a Gaussian distribution $N(v_{\phi_i}, \sigma)$, where $v_{\phi_i}$ is the mean centered at the LLM agent's inferred parameter values for $\phi_i$, and $\sigma$ is the user-defined variance. The execution of $\pi_{o_i}$ is also influenced by the initial states of $o_i$, which are determined by both the skill policy $\pi_{o_{i-1}}(a|s; \phi_{i-1})$ and the termination condition $\beta_{o_{i-1}}(\varphi_{i-1})$ of its preceding skill $o_{k-1}$ in the skill chain. Thus, the training for $\pi_{o_i}(a|s\ \phi_i)$ also needs to optimize the parameters associated with $o_{i-1}$, which involves finding the correct policy parameters for $\phi_{i-1}$ and properly setting its termination condition parameters $\varphi_{i-1}$:

$$\max_{\phi_{i-1}, \varphi_{i-1}, \pi_{o_i}} \mathbb{E}_{s_0 \sim \eta_{o_i}[\phi_{i-1}, \varphi_{i-1}], \phi_i \sim \tilde{q}_{\phi_i}, \tau \sim \pi_{o_i}(a_t|s_t;\phi_i)} \left[ \sum_{t=0}^{T} \gamma^t \mathcal{R}_{o_i}[\phi_i](s_t, a_t) \right] \quad (1)$$

where $\eta_{o_i}$ is the initial state distribution of $o_i$.

A key choice L2S makes is what initial state distribution $\eta_{o_i}$ to choose to train the skill policy $\pi_{o_i}$. Consider a prefix of a skill chain $\rho_k = o_0 \circ o_1 \circ \ldots \circ o_{k-1}$, where all policies for the skills $\pi_{o_0}$ through $\pi_{o_{k-1}}$ along the chain have been trained. L2S chooses the initial state distribution $\eta_{o_k} = \eta_{\rho_k}$ for training $\pi_{o_k}$ to be the distribution of states reached by the controller $\pi_{\rho_k}$ (Sec. 2) from a random environment initial state $s_0 \sim \eta$. The induced distribution $\eta_{\rho_k}$ is defined inductively on the length of $\rho_k$. Formally, for the zero-length path $\rho_k$ (so $\pi_{o_k} = \pi_{o_0}$), we define $\eta_{\rho_k} = \eta$ to be the initial state distribution of the MDP $e$. Otherwise, we have $\rho_k = \rho_{k-1} \circ \pi_{o_{k-1}}$. Then, we define $\eta_{\rho_k}$ to be the state distribution over $\beta_{o_{k-1}}$ (the termination condition of $o_{k-1}$) induced by any trajectory $\tau$ generated using $\pi_{o_{k-1}}$ from $s_0 \sim \eta_{\rho_{k-1}}$. Given an infinite trajectory $\tau = s_0 \to s_1 \to \ldots$ if there exists $i$ such that $\beta_o(s_i)$ holds, we denote the smallest such $i$ by $i(\tau, \beta_o)$. Formally, $\eta_{\rho_k}$ is the probability distribution over $\beta_{o_{k-1}}$ such that for any set of states $S' \subseteq \beta_{o_{k-1}}$, the probability of $S'$ according to $\eta_{\rho_k}$ is

$$\Pr_{s \sim \eta_{\rho_k}[\phi_{k-1}, \varphi_{k-1}]} \left[ s \in S' \right] = \Pr_{s_0 \sim \eta_{\rho_{k-1}}, \tau \sim \pi_{o_{k-1}}(a_t|s_t; \phi_{k-1})} \left[ s_{i(\tau, \beta_{o_{k-1}}[\varphi_{k-1}])} \in S' \right].$$

We note that $\eta_{\rho_k}$ is conditioned on the policy parameters $\phi_{o_{k-1}}$ of the skill policy $\pi_{o_{k-1}}(\cdot|\cdot; \phi_{o_{k-1}})$ and the parameters $\varphi_{k-1}$ of the termination condition $\beta_{o_{k-1}}[\varphi_{k-1}]$, while $\eta_{\rho_{k-1}}$ is unconditioned because the training of $\pi_{o_{k-1}}$ must have already optimized the parameters of skill $o_{k-2}$ (for $k \geq 2$).

**Main Algorithm**. We depict the overall skill training algorithm of L2S in Algorithm 1. It handles a sequence of tasks $\mathcal{T} = \{(e, \mathcal{L}_e)\}$ each with task MDP $e$ and text description $\mathcal{L}_e$. At line 3, it prompts

---

**Algorithm 1** L2S LearningAlgorithm

---

**Require:** A sequence of tasks $\mathcal{T} = \{(e, \mathcal{L}_e)\}$ each with task MDP $e$ and text description $\mathcal{L}_e$
**Require:** Code generating LLM LLMAgent
**Ensure:** Skill Library $\mathcal{O}$, Task Controllers $\mathcal{C}$
1: $\mathcal{O} \leftarrow \varnothing, \mathcal{C} \leftarrow \varnothing$
2: **for each** task $(e, \mathcal{L}_e) \in \mathcal{T}$ **do**
3: $\quad (\rho \equiv o_0 \circ o_1 \circ \dots), \mathcal{R}_o, \beta_o, \mathcal{L}_o \leftarrow$ LLMAgent(prompt(encode($e$), $\mathcal{L}_e, \mathcal{O}$))
4: $\quad$ **for** $k = 0, 1, \dots, \text{LEN}(\rho) - 1$ **do**
5: $\quad\quad$ Train $\pi_{o_k}$ and update the policy parameters $\phi_{k-1}$ and the termination condition parameters
$\quad\quad \varphi_{k-1}$ for the preceding skill $o_{k-1}$ (when $k > 1$) based on Equation 1
6: $\quad\quad \mathcal{O} \leftarrow \mathcal{O} \cup \{o_k, \mathcal{L}_{o_k}\}$
7: $\quad \triangleright$ Optimize the parameters of the last skill $o_{\text{LEN}(\rho)-1}$ using the sparse reward function $R_e$ in $e$
8: $\quad k \leftarrow \text{LEN}(\rho)$
9: $\quad \phi_{k-1}, \varphi_{k-1} \leftarrow \arg\max_{\phi_{k-1}, \varphi_{k-1}} \mathbb{E}_{s \sim \eta_{\rho_k}[\phi_{k-1}, \varphi_{k-1}]}[R_e(s, \pi_{o_{k-1}}(s))]$
10: $\quad \pi_\rho \leftarrow (\pi_{o_0}[\phi_0], \beta_{o_0}[\varphi_0]) \circ \dots \circ (\pi_{o_{k-1}}[\phi_{k-1}], \beta_{o_{k-1}}[\varphi_{k-1}]) \quad \triangleright$ Skill chaining policy for $e$
11: $\quad \mathcal{C} \leftarrow \mathcal{C} \cup \{\pi_\rho\}$

---

the LLM agent with the pythonic representation of $e$, $\mathcal{L}_e$ and the skill library $\mathcal{O}$ (initialized to empty) to generate the skill chain $\rho$ for $e$ (Sec. 3.1). For each task, at line 5, it iteratively trains the skills in $\rho$ (Sec. 3.2). When the LLM agent selects a skill $o_k$ from the skill library $\mathcal{O}$, the algorithm trains the skill controller $\pi_{o_k}$, beginning with the existing policy and value functions (and the replay buffer if using an offline RL algorithm), which often leads to policy reuse or results in fast convergence. At line 6, the algorithm incorporates the trained skill into the skill library $\mathcal{O}$ for reuse in future tasks. At line 9, it optimizes the parameters of the last skill in the skill chain $\rho$ using the *sparse* environment reward $R_e$ from $e$. The final skill chaining controller $\pi_\rho$, constructed for $\rho$ (line 10), is added to $\mathcal{C}$, which holds the controllers for all the tasks in the input sequence $\mathcal{T}$ (line 11).

### 3.3 REINFORCEMENT LEARNING FOR SINGLE SKILLS

We now describe how L2S learns a policy $\pi_{o_k}$ for a single skill $o_k$ based on Equation 1 once the initial state distribution $\eta_{o_k} = \eta_{\rho_k}$ is known (Line 5 of Algorithm 1). At a high level, it trains $\pi_{o_k}$ based on the reward function $\mathcal{R}_{o_k}(\phi_k)$ with the parameters $\phi_k \sim \tilde{q}_{\phi_k}$ sampled from a distribution $\tilde{q}_{\phi_k}$ (akin to "goals" in goal-conditioned reinforcement learning). Specifically, it uses Equation 2 to optimize the parameters of the preceding skill based on (freezed) $\pi_{o_k}$, which can be solved using any black-box optimization algorithms such as CEM.

$$\max_{\phi_{k-1}, \varphi_{k-1}} \mathbb{E}_{s_0 \sim \eta_{\rho_k}[\phi_{k-1}, \varphi_{k-1}], \phi_k \sim \tilde{q}_{\phi_k}, \tau \sim \pi_{o_k}(a|s;\phi_k)} \left[ \sum_{t=0}^{T} \gamma^t \mathcal{R}_{o_k}[\phi_k](s_t, a_t) \right] \quad (2)$$

It uses Equation 3 to learn $\pi_{o_k}$ based on the parameters of its preceding skill, which can be solved using a standard RL algorithm such as SAC (Haarnoja et al., 2018).

$$\max_{\pi_{o_k}} \mathbb{E}_{s_0 \sim \eta_{\rho_k}[\phi_{k-1}, \varphi_{k-1}], \phi_k \sim \tilde{q}_{\phi_k}, \tau \sim \pi_{o_k}(a|s;\phi_k)} \left[ \sum_{t=0}^{T} \gamma^t \mathcal{R}_{o_k}[\phi_k](s_t, a_t) \right] \quad (3)$$

Our skill training algorithm iteratively optimizes both Equation 2 and Equation 3 until convergence.

## 4 EXPERIMENTS AND EVALUATION

**Benchmarks.** We demonstrate the capability of L2S across various environments and tasks within the Meta-World Yu et al. (2019) and ManiSkill2 Gu et al. (2023) benchmarks. Meta-World is an open-source simulated benchmark designed for meta-reinforcement learning and multi-task learning. We conducted tasks within the LORL-Meta-World environment Nair et al. (2021) (Fig. 3 left), a simulated domain built atop Meta-World. This environment features a Sawyer robot interacting with a tabletop setup that includes a drawer, a faucet, and two mugs. As detailed in Table 1 left, we evaluated five tasks: Open drawer, Turn faucet left, Turn faucet right, Push white mug backward, and Push white mug left. Additionally, we introduced a multi-goal task that require a combination of two basic tasks (Task 6).

| LORL-Meta-World Task Sequence | ManiSkill2 Task Sequence |
|---|---|
| Task1: Open drawer | Task1: OpenDrawer |
| Task2: Turn faucet left | Task2: CloseDrawer |
| Task3: Turn faucet right | Task3: PickCube |
| Task4: Push white mug backward | Task4: StackCube |
| Task5: Push white mug left | Task5: PlaceCubeDrawer |
| Task6: Turn faucet left and Open drawer | Task6: OpenDrawer, PlaceCubeDrawer and CloseDrawer |

Table 1: Descriptions of tasks in the environments shown in Fig. 3. The left table outlines the sequence of tasks executed in the LORL-Meta-World, while the right table details the task sequence for ManiSkill2.

ManiSkill2 offers a diverse range of simulated object manipulation tasks. We integrate the cube and cabinet environments in ManiSkill2 (Fig. 3). Sawyer robots in this environment can interact with two cubes and a cabinet having drawers and doors. We evaluated five basic tasks, as summarized in Table 1 (right). We also introduced a multi-goal task (Task 6) that requires the robot to open the cabinet drawer, place a cube inside, and then close the drawer.

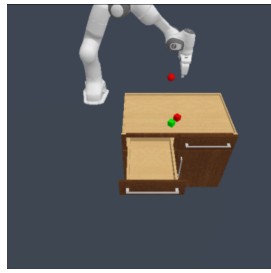

(a) LORL-Meta-World          (b) ManiSkill2

Figure 3: Benchmark Environments

The full list of evaluated tasks and their corresponding instructions can be found in Appendix D. Detailed prompt examples can be found in the Appendix G.

**Baselines.** We conducted a comparative analysis between L2S and two other state-of-the-art methods: Text2Reward (T2R) Xie et al. (2023) and Eureka Ma et al. (2023). **T2R** utilizes LLMs to generate dense reward functions for training a single, monolithic policy per robotic task, using the same Python-based environment abstraction and task description as ours provided to the LLM. In contrast, **Eureka** employs an evolutionary approach, where it inputs the environment script into the LLM to generate multiple reward functions simultaneously for training policies in parallel. Batch success rates are then used to guide the LLM in refining reward functions for the next iteration, creating a feedback loop that iteratively improves the reward functions. L2S differs from T2R in its ability to generate reusable skills for *sequential task learning* while iteratively refining the parameterization of reward and termination functions based on skill chain training. While Eureka's evolutionary approach can adjust reward functions, it relies heavily on costly LLM interactions for environment feedback and expensive policy training with each parameter update, and it lacks support for skill learning. For our benchmarks, we ran Eureka for 3 rounds with 8 samples per round. This process resulted in significantly higher training costs compared to L2S, measured by the environment steps required for agent training. For Eureka, we conducted multiple runs and reported results only from those that had at least one successful sample in each round.

**Ablation.** We also included a variant of L2S called L2S-fixed, which uses fixed LLM-generated parameters in reward and termination functions, instead of optimizing them as in L2S , to assess the impact of addressing potentially incorrect parameter choices made by LLMs.

**Experiment setup.** We use GPT-4 as our LLMAgent. For reinforcement learning of skill policies, we employ Soft Actor-Critic (SAC, Haarnoja et al. (2018)) algorithms, maintaining consistent hyperparameters across all tasks and experiments within these benchmarks. To evaluate the robustness of L2S , each task was conducted using 5 different random seeds. The hyperparameters for SAC are detailed in Appendix C.

**Overall Results.** Fig. 4 illustrates the training results, showing the number of tasks that have converged as the total training timesteps increase. Fig. 5 displays the average evaluation success rates at convergence across all tasks. For the performance on the task sequence of 6 tasks, as shown in Fig. 4: 1) In LORL, L2S solved an average of 5.44 tasks in a total of 1.1e7 time steps, while L2S-fixed solved 4.98 tasks, and Text2Reward solved 4.9 tasks in a total of 1.5e7 time steps. 2) In

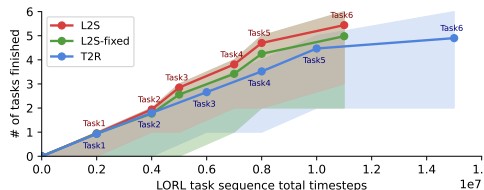 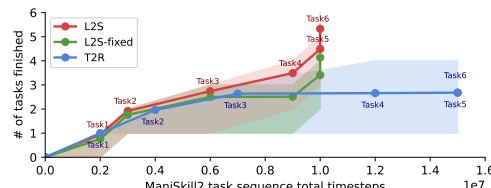

Figure 4: Given the sequence of tasks in Table 1, we report the average number of tasks trained to convergence on LORL-Meta-World (left side) and ManiSkill2 (right side), averaged over 5 random seeds. The policy is considered converged when its evaluation success rate converges to a value significantly above zero. Eureka is omitted from here because its evolutionary reward function search demands considerably more training steps than the other methods.

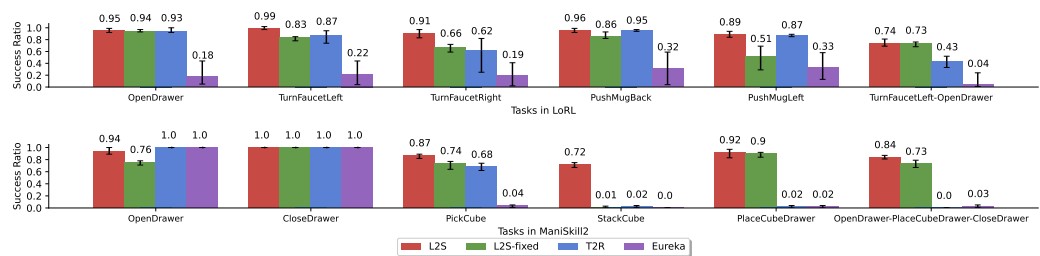

Figure 5: We report the average evaluation success rates at convergence across all tasks on LORL-Meta-World (top) and ManiSkill2 (bottom), averaged over 5 random seeds.

ManiSkill2, L2S solved an average of 5.33 tasks in a total of 1e7 time steps, while L2S-fixed solved 4.14 tasks, and Text2Reward solved only 2.68 tasks in a total of 1.5e7 time steps. Overall, L2S showed an improvement of 11.0% in LORL and 98.8% in ManiSkill2, while requiring 26.7% and 33.3% less training cost compared to the baseline Text2Reward. For the performance on single simple or complex tasks, as shown in Fig. 5, L2S outperformed the baseline Text2Reward by 18.7% and 98.7% on average success rate in LORL and ManiSkill2 environments, respectively, demonstrating a significant performance improvement with L2S. Additional experiment results can be found in Appendix E.

**Skill Reusing.** Specifically, in the LORL environment, L2S leverages skills learned from "Turn faucet left" and "Push white mug backward" to expedite training for "Turn faucet right" and "Push white mug left" respectively. As shown in Fig. 6, the first skill of the "Turn faucet left" task, $\pi_{o_1}(a|s; \texttt{params})$, guides the robot's end-effector to the right side of the faucet handle at a target location with $\texttt{params[0]} = 0.083$m away from the handle. This parameter-conditioned skill was reused with $\texttt{params[0]} = -0.095$m to guide the end-effector to the opposite side of the faucet in the "Turn faucet right" task. In Task 6 of the LORL environment, although the LLM agent recognizes that this combination of tasks can be ad-

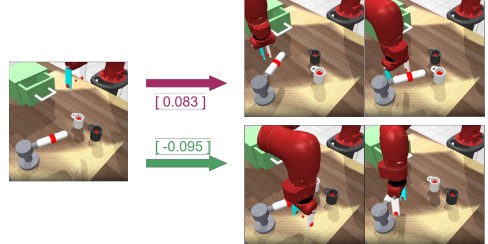

Figure 6: An example of skill reusing. The first skill $\pi_{o_1}(a|s; \texttt{params})$ in task "Turn faucet left" is reused for "Turn faucet right" with the parameter value $\texttt{params}$ optimized from [0.083] to [-0.095].

dressed by reusing existing skills, L2S still requires several training steps to fine-tune these skills for adaptation to the environment due to shifts in the initial state distributions. Similarly, in the ManiSkill2 environment, L2S leverages the skill for approaching the handle in the "Open Drawer" task for the "Close Drawer" task. It also reuses the skill for grasping the cube in the "Pick Cube" task for the "Stack Cube" and "Place Cube Drawer" tasks, thereby expediting sequential task learning.

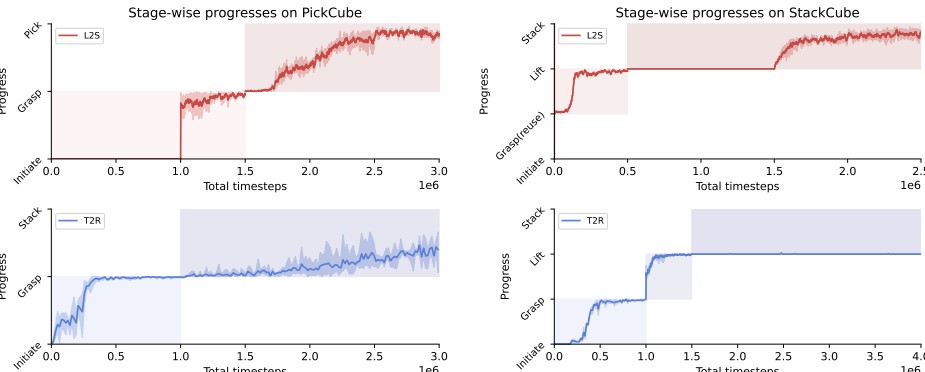

Figure 7: Left: the progress of training the task "PickCube" in ManiSkill2 measured by 2 stage-wise evaluation functions: **(1)grasp cube and (2) place cube**. Right: the progress of training the task "Stackcube" in ManiSkill2 measured by 3 stage-wise evaluation functions: **(1)grasp cube, (2) lift cube, (3) stack cube**. In sequential task learning, L2S reuses the skills for grasping cubes from "PickCube" in training "StackCube".

**Training progress evaluation.** As summarized in Fig. 5, L2S significantly outperforms the baseline T2R on challenging tasks like Stack Cube. To further illustrate this performance, we present the training curves of both L2S and T2R in Fig. 7 for the PickCube and StackCube tasks. We utilized functions from the ManiSkill2 library to design evaluation functions that assess the progress of the learning agent in achieving specific key subgoals of each task. In the case of StackCube, the evaluation function gauges whether the agent has consistently mastered the abilities to grasp, lift, and stack a cube. Although T2R can generate step-wise reward functions, combining rewards across different steps proves to be difficult. In the case of PickCube or StackCube, a high grasping reward combined with a relatively lower stacking reward leads the policy to prioritize holding the cube, as ineffective stacking actions can easily result in losing contact with the cube, thus yielding a lower reward. T2R also requires significantly more training steps than L2S to achieve convergence (Fig. 4), as it learns a single monolithic policy that lacks the flexibility for easy reuse. In contrast, Fig. 7 demonstrates that L2S quickly acquires the ability to grasp a cube in StackCube by effectively reusing the grasp skill learned during the PickCube task. Eureka faces similar challenges as T2R in generating effective step-wise reward functions, with performance declining as the complexity of the required reward functions increases.

**Ablation Study.** In Fig. 4, although the ablation L2S-fixed demonstrates a similar convergence rate to L2S for tasks in the LORL-Meta-World environment, Fig. 5 reveals that it converges to sub-optimal policies compared to L2S . In the ManiSkill environment, L2S-fixed struggles to solve more challenging tasks, such as StackCube, underscoring the necessity of optimizing LLM-generated parameters in the reward and termination functions to address the inherent uncertainty of LLM agents when dealing with low-level environmental control intricacies.

## 5 RELATED WORK

**LLM Planning for Robotics.** Recent research has highlighted the integration of Large Language Models (LLMs) in robotic task and motion planning (TAMP) (Firoozi et al., 2023). Huang et al. (2022a) investigated LLMs for direct trajectory planning, revealing limitations in spatial and numerical reasoning that necessitate frequent re-prompting to align with task constraints. Following works aim to mitigate the gap on feasibility and correctness when applying LLM-generated plans to simulated or real-world robotic environments. Inspired by the in-context learning ability of LLMs, **Inner Monologue** (Huang et al., 2022b) allows robotic systems to integrate real-time environmental feedback into LLM-generated plans. This strategy significantly enhances the adaptability and effectiveness of robotic agents by using continuous feedback to adjust planning strategies. **Text2Motion** (Lin et al., 2023) goes a step further by not only generating feasible task plans (a sequence of skills) but also ensuring these plans are geometrically executable before initiation. Another direction is to

utilize LLMs for translating natural language into intermediate formal task representations, **NL2TL** (Chen et al., 2024a) and **AutoTAMP** (Chen et al., 2024b) significantly enhancing task completion through auto-regressive error correction of both syntax and semantics. The planning ability of LLMs plays a great role in L2S for generating skill chains and make plan on it to complete robotic tasks. To finish wider range of tasks, **BOSS**(Zhang et al., 2023) leverages LLM to build skill library with large amount of complex and useful skill chains generated from a set of primitive skills. Also, **SayCan** (Ahn et al., 2022) ranks all the possible skills by the task-grounding probability (usefulness) and world-grounding probability (feasibility) and select the one with highest probability at each step for LLM decision making within a given embodiment.

**LLM-Based Code Generation. L2R** (Yu et al., 2023) introduces a new paradigm that harnesses flexibility of reward function representations by utilizing LLMs to define reward parameters that can be optimized and accomplish variety of robotic tasks. To generate RL reward function for robotics tasks, Zeng et al. (2024) includes self-align ranking to improve the quality of generated reward function using samples ranked by both LLMs and reward function. **Text2Reward** (Xie et al., 2023) generates interpretable, free-form dense reward functions as an executable program grounded in a compact representation of the environment either by zero-shot or few-shot. **Eureka** (Ma et al., 2023) generates dense reward function without any task-specific prompting or pre-defined reward templates (zero-shot). Both **Text2Reward** and **Eureka** leverage LLM's in-context ability to improve reward function by providing human-involved feedback or automated feedback, respectively. In **League++** (Li et al., 2024), the reward functions are generated by the LLM through selecting and weighting pre-defined metric functions provided by human experts. Our method differs from it in two key aspects:1)Free-form Reward Generation and 2)Reduced Human Expert Effort. Another two studies explores using LLMs for robot-centric policy generation, termed **ProgPrompt** (Singh et al., 2022) and **Code as Policies** (Liang et al., 2023), which involves generating control code directly from language instructions. Our L2S differs from the aforementioned works in several ways: 1) it decomposes tasks into chain of skills, 2) it learns skills as primitives and building skill library based on the skill chains, 3) it optimizes parameters in generated functions to enhance task performance, and 4) it reuses skills and skill chains from the library to improve learning efficiency.

Due to the limitation of pages, more related work can be found in Appendix A.

## 6 LIMITATIONS

L2S has been evaluated solely in robotic manipulation domains. Applying LLM-based skill discovery to other task types, such as navigation, would necessitate more advanced reasoning about environmental structures, which we leave as an avenue for future research. Additionally, L2S currently operates within state-based environments, as the LLM-generated termination functions require explicit state information to assess whether termination thresholds are met. Extending this approach to vision-based tasks may require training a supervised model that learns from state-based termination conditions, an aspect we plan to explore in future work. As for the assumption that each skill reply on the performance of the skill before, to let the framework figure out which skills need to be optimized might be a great extension. Lastly, LLM hallucinations present challenges in generating robust free-form reward and termination function code. Constraining code generation within a structured intermediate representation, possibly defined by a domain-specific language, might offer a balance between generation stability and the exploration of the reward space.

## 7 CONCLUSION

We present L2S that leverages Large Language Models (LLMs) to autonomously construct a skill library for sequential task learning. L2S progressively builds a skill library guided by LLMs and efficiently reuses them across new tasks, enabling the learning algorithm to effectively handle increasingly challenging environments. To handle the uncertainty in LLM-generated reward and termination functions, L2S trains a parameter-conditioned policy that perform well across a broad range of parameter values for each skill and selects the most suitable parameter values during the training of its subsequent skills, mitigating the risk of incorrect parameter choices by LLMs. Experimental results demonstrate that L2S outperforms baselines in solving complex, multi-step tasks, largely due to its ability to automatically construct a skill library for sequential task learning.

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

## A    MORE RELATED WORK

**LLM for Reasoning.** Recent studies on large language models (LLMs), e.g. **GPT-3** (Brown et al., 2020), **GPT-4** (OpenAI et al., 2024), have demonstrated significant advancements of its reasoning capabilities. Prompting proposed by **Chain-of-Thought (CoT)** (Wei et al., 2023), has shown efficacy in improving reasoning by eliciting detailed reasoning paths in LLMs, which helps in tasks involving multi-step reasoning. Similarly, **ReAct** (Yao et al., 2023) combines reasoning with actions, enhancing performance on tasks by enabling dynamic reasoning and interactions with external information, demonstrating significant improvements. Additionally, **Zero-shot-CoT** (Kojima et al., 2023) has been proved effective in enhancing the zero-shot reasoning abilities of models across various tasks, enhancing the potential of LLMs in tasks requiring complex multi-hop thinking without the need for task-specific fine-tuning. These advancements suggests a promising direction for further enhancing the reasoning powers of LLMs through advanced prompting techniques and integrated reasoning-action paradigms. We leverage such reasoning ability to make LLMs understand the semantics of robotic tasks and follow the instructions from human correctly.

## B    DISCUSSION

Our tool demonstrates that decomposing natural language tasks into skill chains significantly enhances performance across a broad range of robotic tasks while reducing the cost of neural policy learning. One associated open problem is the instability caused by the hallucinations of LLMs, which can lead to unreliable code generation. Without fine-tuning LLMs, effective methods to address this issue include reducing the randomness of the next token generated by LLMs or iteratively sampling code via feedback prompts. The reward and termination functions can principally be used by RL agents to train control policies using visual inputs and to terminate when specific visual conditions satisfy the termination criteria. Extending L2S to vision-based environments is left for future work. We hope our work represents a meaningful effort to apply LLMs to code generation across various research domains, not limited to reinforcement learning, and contributes valuable insights to the development of this field.

## C    HYPER-PARAMETERS

We document the hyper-parameters used for LLM code generation and RL learning algorithms in this section. For generating dense reward function code and termination condition function code, we utilized GPT-4 as the LLM agent with the sampling temperature set to 0.2 and the top_p (the cumulative probability of next token candidates) set to 0.1 for each experiment in L2S . The baseline (T2R) maintained the default values for temperature and top_p at 0.7 and 1, respectively.

For the reinforcement learning algorithm, we employed the implementation from Stable-Baselines3 (Raffin et al. (2021)) with the hyper-parameters listed in Table 2.

Table 2: Hyper-parameter of SAC algorithm applied to tasks in two benchmarks

| SAC Hyper-parameters | LORL(Meta-World) | ManiSkill2 |
|---|---|---|
| Discount factor $\gamma$ | 0.99 | 0.95 |
| Target update frequency | 2 | 1 |
| Learning rate | $3e^{-4}$ | $3e^{-4}$ |
| Train frequency | 1 | 8 |
| Soft update $\tau$ | $5e^{-3}$ | $5e^{-3}$ |
| Gradient steps | 1 | 4 |
| Learning starts | 4000 | 4000 |
| Hidden units per layer | 256 | 256 |
| # of layers | 3 | 2 |
| Batch Size | 512 | 1024 |
| Initial temperature | 0.1 | 0.2 |
| Rollout steps per episode | 500 | 100/200 |
| Replaybuffer size | 5e5 | 5e5 |

# D  TASK LIST

In this section, we list all tasks examined in both LORL and ManiSkill2 benchmarks separately in Table. 3 and Table. 4, accompanied by their corresponding natural language instructions. Note that these instructions constitute part of the task prompt explicitly.

Table 3: List of tasks in LORL

| Single-goal Task | Instruction |
|---|---|
| Push mug backward | Move the white mug backward 0.1 meter. |
| Push mug left | Move the white mug left 0.1 meter. |
| Turn faucet left | Turn the faucet handle left $\frac{\pi}{4}$ radian distance. |
| Turn faucet right | Turn the faucet handle right $\frac{\pi}{4}$ radian distance. |
| Open drawer | Open the drawer until the position of drawer box is greater than target value. |
| **Multi-goal Task** | |
| Push mug backward and open drawer. | |
| Open drawer and turn faucet left. | |
| Push mug backward and turn faucet left. | |
| Push mug backward and open drawer and turn faucet left. | |

Table 4: List of tasks in ManiSkill2.

| Task | Instruction |
|---|---|
| Pick cube | Pick up cube A, move it to goal position and hold it. |
| Stack cube | Pick up cube A and place it on top of cube B. |
| Open cabinet drawer | A single-arm mobile robot needs to open a cabinet drawer. |
| Close cabinet drawer | A single-arm mobile robot needs to close a cabinet drawer. |
| Open drawer, Place cube and close drawer | Open the cabinet drawer, place cube it into the drawer and close the drawer. |

# E  ADDITIONAL EXPERIMENT RESULTS

In this section, we show the results of:

- The error analysis on LLM-generated functions.
- Optimizing function parameters with different parameter variances.
- Performance of L2S on more long-horizon tasks.

## E.1  ERROR ANALYSIS ON GENERATED FUNCTIONS.

For the reward generation experiment in the LORL and ManiSkill2 environments, we selected 5 simple tasks from each environment(LORL: "OpenDrawer, TurnFaucetLeft, TurnFaucetRight, PushMugBack, PushMugLeft"; ManiSkill2: "OpenDrawer, CloseDrawer, PickCube, StackCube, PlaceCubeDrawer"), as shown in Figure 5 in the paper, and queried the LLM for 20 samples per task. Across these 10 tasks, the number of skills generated ranged from 2 to 5. The reported results reflect the success rate for completing the entire tasks.

As shown in Table 5, L2S achieves a higher execution success rate for each generated skill compared to the whole-task reward function generated in Text2Reward. This is because generating free-form function code for individual skills is inherently simpler than generating a single function for the entire task. Each skill represents only a portion of the overall task, reducing complexity.

Table 5: Error Analysis on generated functions. We evaluated the LLM's performance in generating correct function code for both LORL and ManiSkill2 environments more than 100 samples each.

| LLM(GPT-4) | LORL | ManiSkill2 |
|---|---|---|
| Correct | 92% | 87% |
| Syntax/Shape Error | 8% | 13% |

The results highlight the effectiveness of L2S in breaking down complex tasks into manageable components and improving the reliability of code generation.

### E.2 OPTIMIZING PARAMETERS WITH DIFFERENT VARIANCES.

As we provided information about the environment and additional knowledge that connects the semantics of real-world instructions to the robot environment and specifies the task's successful conditions (see Appendix E), the LLM gains some understanding of the environment's scale and selects reasonable (though not necessarily optimal) parameter mean values. By default, we set the variance of the parameters to be twice the maximum mean value generated by the LLM for the current task (a heuristic). We conducted experiments with varying alternative parameter value variances, while keeping the parameter mean value fixed. The results were obtained using three different random seeds and demonstrate that L2S consistently achieves optimal parameter values across the default setting and all tested variants.

#### E.2.1 LORL-TURN FAUCET LEFT

For this task, we examined three different parameter variance combinations—variants 1, 2, and 3—to analyze their effects on reward and termination parameters. The first skill in the task is trained to position the end effector around the faucet handle, with the reward function parameter defining the acceptable distance to the handle. The termination condition parameter specifies how close the end effector must be to the target position to transition to the next skill. Results are shown in Table 6.

- **Variant 1**: The variance of the termination condition parameter is increased.
- **Variant 2**: The variance of the reward function parameter is increased.
- **Variant 3**: The variance of both parameters is increased.

Table 6: Optimizing parameters with different variances in LORL.

| Reward Func Parameters / Termination Func Parameters | Initial Parameters Mean Value | Initial Parameters Variance | Optimized Params/(Std) | Success Rate/(std) | Training Cost (Timesteps) |
|---|---|---|---|---|---|
| Default | [0.01]/[0.01] | [0.2]/[0.02] | [0.107/(0.02)]/[0.013/(0.001)] | 0.99/(0.01) | 1e6 |
| **Variant1** | [0.01]/[0.01] | [0.2]/**[0.05]** | [0.118/(0.04)]/[0.027/(0.005)] | 0.89/(0.13) | 1.5e6 |
| **Variant2** | [0.01]/[0.01] | **[0.5]**/[0.02] | [0.16/(0.01)]/[0.015/(0.0003)] | 0.97/(0.04) | 1e6 |
| **Variant3** | [0.01]/[0.01] | **[0.5]**/**[0.05]** | [0.114/(0.01)]/[0.031/(0.001)] | 0.93/(0.05) | 1e6 |

### E.3 MANISKILL2-OPEN DRAWER

For this task, the parameter in the termination condition of the first skill specifies the required proximity of the robot's end effector to the target position above the drawer handle. In the variant, we increase the variance of this parameter from the default value of 0.02 to 0.05 to evaluate its impact on performance. Results are shown in Table 7.

Table 7: Optimizing parameters with different variances in ManiSkill2.

| Reward Func Parameters | Initial Parameters Mean Value | Initial Parameters Variance | Optimized Params/(Std) | Success Rate/(std) | Training Cost (Timesteps) |
|---|---|---|---|---|---|
| Default | [0.01] | [0.02] | [0.026(0.003)] | 0.94/(0.06) | 1e6 |
| **Variant** | [0.01] | **[0.05]** | [0.031(0.006)] | 0.97/(0.02) | 1.5e6 |

### E.3.1 Long-horizon tasks

We report results on complex, meaningful tasks in both the LORL and ManiSkill2 benchmarks in Table 8. Notably, we prompted GPT-4 in both L2S and Text2Reward to reuse policies learned from prior single tasks whenever possible, ensuring a fair comparison between the two approaches.

Table 8: Performance of L2S on more long-horizon tasks.

| Benchmark | Task | Text2Reward(Std) | L2S(Std) |
|---|---|---|---|
| LORL | PushMugBack-OpenDrawer | 0.91(0.043) | 0.93(0.030) |
| | OpenDrawer-TurnFaucetRight | 0.89(0.096) | 0.93(0.062) |
| | MugBack-OpenDrawer-TurnFaucetRight | 0.76(0.071) | 0.90(0.044) |
| ManiSkill2 | OpenDrawer-Place**TwoCubes**Drawer-CloseDrawer | 0.01(0.002) | 0.72(0.056) |
| | OpenDrawer-Place**ThreeCubes**Drawer-CloseDrawer | 0.01(0.001) | 0.54(0.032) |

## F  Skill reuse and refinement on complex task

In this section, we show the result of reusing skills or skill chain from basic single-goal task to complete complex tasks in Fig. 8. We showcases the effectiveness of skills refinement in L2S when necessary. For example, the Task4 "Turn faucet left and open drawer" performs only 12% success ratio with directly reusing skills from skill library. However, with refinement by L2S , the performance can be greatly improved to close to perfect, with evaluation curve shown in Fig. 9.

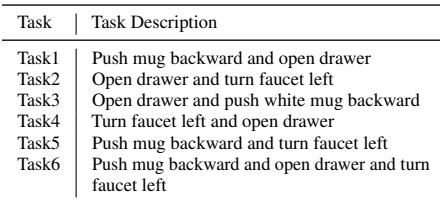
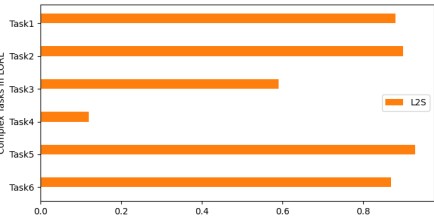

| Task | Task Description |
|---|---|
| Task1 | Push mug backward and open drawer |
| Task2 | Open drawer and turn faucet left |
| Task3 | Open drawer and push white mug backward |
| Task4 | Turn faucet left and open drawer |
| Task5 | Push mug backward and turn faucet left |
| Task6 | Push mug backward and open drawer and turn faucet left |

Figure 8: Complex task instruction(left) and success ratio on complex tasks in LORL environment(right).

## G  LLM prompt

A prompt used in L2S consists of following components: *introduction*, *environment description*, *additional environmental knowledge*, *tips and tricks*, *instruction hint*, and *learned skill library*. Here we use an example of the prompt for ManiSKill2 manipulation tasks to demonstrate how each component is formatted:

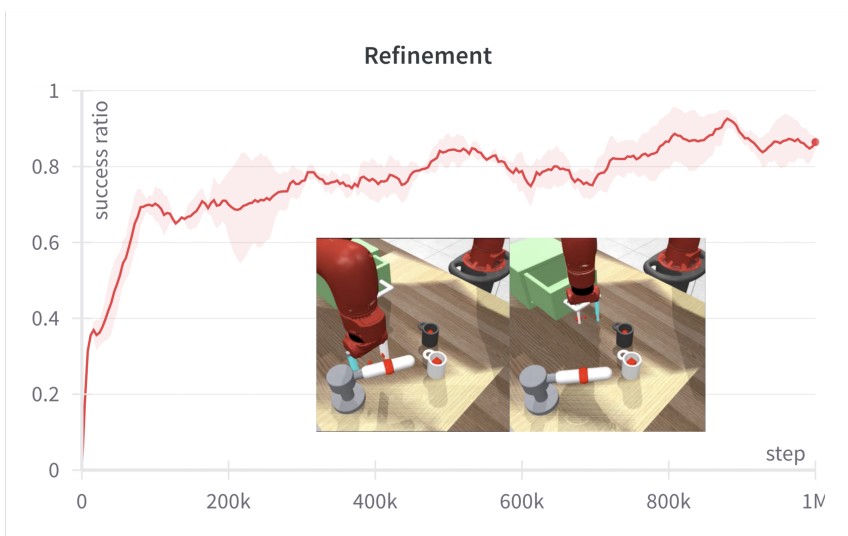

Figure 9: Success ratio on refining reused skill in task "Turn faucet left and open drawer".

**Listing 1** Introduction.

```
introduction = """
You are an expert in robotics, reinforcement learning, task decomposing, task planning and code
generation. We are going to control a robotic arm to complete some given tasks. The robotic arm is a
7-DoF Fetch Mobile Manipulator with a two-fingered parallel gripper. The robotic arm is controlled by
small displacements of the gripper in Cartesian coordinates and the inverse kinematics are computed
internally by the MuJoCo framework. The gripper can be opened or closed in order to perform the
grasping operation of pick and place.

The action space of the robot is Box(np.array([-1, -1, -1, -np.pi, -1]), np.array([1, 1, 1, np.pi, 1]),).

| Num  | Action                                                              | Unit         |
| ---- | ------------------------------------------------------------------- | ------------ |
| 0    | Displacement of the end effector in the x direction dx              | position (m) |
| 1    | Displacement of the end effector in the y direction dy              | position (m) |
| 2    | Displacement of the end effector in the z direction dz              | position (m) |
| 3    | Angular displacement of the end effector                            | position (m) |
| 4    | Positional displacement per timestep of each finger of the gripper| position (m) |

Now I want you to help me
1) decompose the robotic task into sequences of skills
2) write dense reward functions and termination conditions of reinforcement learning for each skill.
3) make plan on the skills to finish the robotic task.

I'll give you the attributes of the environment and robotic arm itself. You can use these class attributes
to write the reward function.
"""
```

**Listing 2** Environment description.

```
environment_description = """
The following classes provide the information about the robotic arm and all objects in the environment.

class BaseEnv(gym.Env):
    self.robot : SawyerRobot # the robot in the environment
    self.white_mug : MugObject # the white mug in the environment
    self.black_mug : MugObject # the black mug in the environment
    self.faucet : FaucetObject # the faucet object in the environment
    self.drawer : DrawerObject # the drawer object in the environment

class SawyerRobot:
    self.ee_position : np.ndarray[(3,)] # indicate the 3D position of the end-effector
    self.gripper_finger_distance : numpy.float64
            # indicate the distance between the gripper fingers away from the initial position
            # range between 0 and 0.1.
            # The closer the grippers, the smaller the value
    self.init_ee_position : np.ndarray[(3,)] # indicate the initial 3D position of the end-effector
    self.init_gripper_finger_distance : numpy.float64
            # indicate the initial distance between the gripper fingers away from the initial position,
            # range between 0 and 0.1

class MugObject:
    self.position : np.ndarray[(3,)] # indicate the 3D position of the rigid object
    self.init_position : np.ndarray[(3,)] # indicate the initial 3D position of the rigid object

class FaucetObject:

    self.faucet_handle_postion : np.ndarray[(3,)] # indicate the 3D position of the handle of faucet
    self.faucet_handle_angular_position : numpy.float64
            # indicate the angular position of the handle with respect to the faucet in radians.
            # Faucet moving clockwise makes this value smaller.
    self.init_faucet_handle_postion : np.ndarray[(3,)]
            # indicate the initial 3D position of the handle of faucet
    self.init_faucet_handle_angular_position : numpy.float64
            # indicate the initial angular position of the handle with respect to the faucet in radians.
            # Faucet moving clockwise makes this value smaller.

class DrawerObject:
    self.box_handle : np.ndarray[(3,)] # indicate the 3D position of the handle of drawer box
    self.drawer_box_position : numpy.float64 # indicate the 1D relative position of the drawer box.
                                    # The position range is between [-0.16, 0] meter.
    self.init_box_handle : np.ndarray[(3,)] # indicate the initial 3D position of the handle of drawer box
    self.init_drawer_box_position : numpy.float64 # indicate the initial 1D relative position of
                                    # the drawer box
"""
```

**Listing 3** Additional environment knowledge.

```
env_additional_knowledge = """
Additional knowledge:
1. For the robotic arm gripper and all the objects in the environment, the direction words in the
following task are defined as:
    1) "Left" or "right" means towards the positive or negative x-axis with respect to the related object
    position, respectively. X-axis is corresponding to the first value in the 3D position with form
    "np.ndarray[(3,)]". For example, x-axis of mug is "mug.position[0]".
    2) "Forward/Front" or "backward/Back" means towards the positive or negative y-axis with respect to
    the reference object position, respectively. Y-axis is corresponding to the second value in the
    3D position with form "np.ndarray[(3,)]". For example, y-axis of a mug is "mug.postion[1]".
    3) "Above" or "below" means towards the positive or negative z-axis with respect to the reference
    position, respectively. Z-axis is corresponding to the third value in the 3D position with
    form "np.ndarray[(3,)]".
    For example, z-axis of mug is "mug.position[2]".
    4) Specially, for the object faucet, "turn left" or "turn right" means turning faucet that increases
    or decreases the faucet handle angular position.
2. In order to compare the replative positions of different items in the environment, including
the robotic arm gripper and all the objects, you must first identify the attributes that represents
the 3D-positions, and then use these attributes for computation. In practice, the relative position
words in the following task are defined as:
    1) "One item is on the left or on the right of the other item" means the item is on the positive or
    negative x-axis direction with respect to the other item, respectively. X-axis is corresponding
    to the first value in the 3D position with form "np.ndarray[(3,)]".
    2) "One item is in front of or at the back of the other item" means the item is on the positive or
    negative y-axis direction with respect to the other item, respectively. Y-axis is corresponding
    to the second value in the 3D position with form "np.ndarray[(3,)]".
    3) "One item is above or below the other item" means the item is on the positive or negative z-axis
    direction with respect to the other item, respectively. Z-axis is corresponding to the third value
    in the 3D position with form "np.ndarray[(3,)]".
3. Tasks about moving mug are considered successful when mug is moved at least 0.1 meter towards the
correct direction compared with the object's initial position.
4. Tasks about turning faucet are considered successful when faucet is turned at least np.pi/4 radian
towards the correct direction compared with the object initial position.
5. Tasks about opening or closing drawer are considered successful when drawer box is fully open or
fully closes. Drawer fully open means drawer box position is smaller than -0.15 meter.
Drawer fully closed means drawer box position is greater than -0.01 meter.
"""
```

**Listing 4** Instruction hint.

```
instruction_hint = """
Task to be fulfilled: {instruction}.
Here is the instruction:
Please think step by step and finish the following requirements one by one in order:
    1. Tell me what does this task mean. If it is a complex task, identify how many simple task you can
    identify.
    2. Decompose a whole task into a set of possible skills and plan on the skills to finish each simple
    task. You can refer to the above examples if provided after the intruction part.
    3. Identify which example you are referring to, if any.
    4. Identify the index of skill that terminate each simple task as you have answered above. Save the
    index of skill in "simple_task_termination_skill = [...]"
    5. For each skill, design a pair of dense reward function and terminition condition function based
    on the purpose of the skill. Write down the pair of functions one by one with the following format:
        1) Make each pair of reward function and termination function a separate python code piece
        "```python ```".
        2) Dense reward function is used in reinforcement learning, here are the requirements:
            a. Create a list "params = [...]" containing extra parameters that never exist for computing
            reward (if any). But you should not include any threshold value, reward term weight
            or attributes that already exist in the above environment information in the list "params",
            e.g, termination threshold value, reward term weight, the position information of any item.
            Make sure every parameter in list "params = [...]" is used in the dense reward function.
            b. Define the reward term one by one and explain the purpose of each reward term as comment.
            c. This function starts with `def compute_dense_reward_skill_NUM(self, action, obs) -> float`.
            It only returns variable `reward : float`.  Replace 'NUM' with the number of skill.
        3) Terminition condition function decides whether the skill is successful, here are the
        requirements:
            a. Copy list "params = [...]" from dense reward function and paste it into the terminition
            condition function. Any value in list "params" shuold not be used as termination threshold.
            b. Create a list "t_params = [...]" containing the value used as termination threshold if the
            corresponding skill is not in "simple_task_termination_skill". Make sure every parameter
            in list "t_params = [...]" is used later in terminition condition function.
            c. Make lists "params = [...]" and "t_params = [...]" don't conflict with each other because
            they are used for different purposes.
            d. This function starts with  `def termination_skill_NUM(self, obs) -> bool`. It only returns
            variable `done : bool`. Value of `done : bool` should be decided before it is returned.
            Replace 'NUM' with the number of skill.
"""
```

**Listing 5** Few-shot example.

```
Instances of Few-shot Examples:
1.Task to be fulfilled: Turn an object with a handle left.
Corresponding skills and sequence of skills for accomplishing the task:
  Skill 1: Align the robot arm end-effector to a 3D position on the right of the object handle with
  some offset.
  Skill 2: Move robot arm end-effector and turn the object handle left.
  The sequence for accomplishing the task could be: Skill 1 -> Skill 2.
2.Task to be fulfilled: In the MuJoCo PickAndPlace environment, pick up a box and move it to the 3D
goal position and hold it there.
Corresponding skills and sequence of skills for accomplishing the task :
  Skill 1. Navigate gripper to the box.
  Skill 2. Grasp the box and move the box to the goal position and hold it.
  The sequence for accomplishing the task could be: Skill 1 -> Skill 2.
```

**Skills library prompt.** A complete L2S prompt is the ordered concatenation of the above components. Additionally, we could also ask the LLM to generate response considering reusing given library of skills (in listing 6). Such a prompt makes it possible for L2S to reuse either skills generated by language model or reference skills given by human experts in the code generation process, thus potentially facilitate the skill discovery and training.

**Listing 6** Skills library prompt.

```
skills_lib_prompt = """
After finishing the job above, I have one more job for you.
Now we have a skills library which store the already trained skills in a list format, each element in the
list mapping  the stored skills number and the discription of the skills.
skills_library=[
    "1":"Navigate to the position on the left side of the faucet handle and keep some distance away
    from it. This skill has paramters as safe distance between gripper and faucet handle that can
    be tuned.",
    "2":"Navigate gripper to the intermediate position on the forward direction of the white mug
    and maintain a safe distance from the white mug.",
    "3":"Move gripper and push the faucet to the right. This skill can not be modified.",
    ]

What you should do is:
    1) First, think the meaning of each skills. You should clarify the following attributes of each skill:
    the objects involved in the skill, the relation between objects and the goal that the skill finally
    should achieve.
    2) Then, with the decomposed skills of current task and your understanding of the skills in skill
    library, please think about which skill in the library do the exact same work as some skill(s) in
    current task and can be reused.
    3) Lastly, Please give back the pair-wise mapping from current skill number to the skill number in
    skills library with python JSON format. Each pair of skill models selected must have the same
    attribute. That's means even if two skills are very similar but not the same, you should not select
    them because it need extra training. And you should explain the reason why you make such pairing.
"""
```

## H  EXAMPLES OF REWARD FUNCTIONS AND TERMINATION CONDITION FUNCTIONS

In this section, we provides example pairs of generated reward functions and termination functions generated by LLM to solve given tasks. Each figure includes one dense reward function and its corresponding termination function to constitute a complete skill. Numbers in the suffix of the function names denote the indices of the skills.

**Turn faucet left.** Fig. 10 and 11 show the generated skills. In this task, the language model proposes a simple yet effective two-stage solution. Specifically, the first skill is responsible for aligning the robot arm end-effector to a 3D position on the right of the faucet handle. Then the second skill moves robot arm end-effector and turn the faucet handle left and finally solve the task.

```python
def compute_dense_reward_0(self, action, obs, params) -> float:
    params = [0.01]
    # Reward term 1: The negative distance between robot's end-effector and the target
    # position on the right of the faucet handle.
    target_position = obs['current_state'][10:13] + np.array([params[0], 0, 0])
    distance = np.linalg.norm(obs['current_state'][:3] - target_position)
    reward = -distance

    # Reward term 2: Regularization term on the action, to encourage smaller actions
    # for smoother movements.
    action_reg = 0.1 * np.linalg.norm(action)
    reward -= action_reg

    return reward

def termination_0(self, obs, params, t_params) -> bool:
    params = [0.01]
    t_params = [0.01]
    # The skill is successful if the robot's end-effector is close
    # enough to the target position
    target_position = obs['current_state'][10:13] + np.array([params[0], 0, 0])
    distance = np.linalg.norm(obs['current_state'][:3] - target_position)
    done = distance < t_params[0]
    return done
```

Figure 10: Skill 0 for turn faucet left.

```python
def compute_dense_reward_1(self, action, obs, params) -> float:
    params = []
    # Reward term 1: The negative difference between current and initial angular
    # position of faucet handle.
    angular_difference = obs['current_state'][13] - obs['initial_state'][13]
    reward = -angular_difference

    # Reward term 2: Regularization term on the action, to encourage smaller
    # actions for smoother movements.
    action_reg = 0.1 * np.linalg.norm(action)
    reward -= action_reg

    return reward

def termination_1(self, obs, params, t_params) -> bool:
    t_params = [0.78]
    # The skill is successful if the faucet handle has been turned more than 0.78 radian
    # to the left
    angular_difference = obs['current_state'][13] - obs['initial_state'][13]
    done = angular_difference < -t_params[0]
    return done
```

Figure 11: Skill 1 for turn faucet left.

**Open drawer.** Fig. 12, 13 and 14 demonstrate the three skills in the open drawer task decomposition. Concretely speaking, the response from LLM is:

*Skill 1: Move the robot arm end-effector to a 3D position above the drawer box handle.*

*Skill 2: Align robot arm end-effector with the drawer box handle.*

*Skill 3: Move robot arm end-effector to open the drawer.*

*The sequence for accomplishing the task could be: Skill 1 -> Skill 2 -> Skill 3.*

```python
def compute_dense_reward_0(self, action, obs, params) -> float:
    params = [0.01]
    # Reward term 1: The negative distance between the robot's end-effector and the
    # target position above the drawer box handle.
    reward_term_1 = -np.linalg.norm(obs['current_state'][:3] - \
        (obs['current_state'][14:17] + np.array([0, 0, params[0]])))
    # Reward term 2: Regularization term on the robot's action.
    reward_term_2 = -np.linalg.norm(action)
    # The total reward is the weighted sum of the reward terms.
    reward = 1 * reward_term_1 + 0.1 * reward_term_2
    return reward

def termination_0(self, obs, params, t_params) -> bool:
    params = [0.01]
    t_params = [0.01]
    # The skill is considered successful if the distance between the robot's
    # end-effector and the target position is less than a threshold.
    done = np.linalg.norm(obs['current_state'][:3] - \
        (obs['current_state'][14:17] + np.array([0, 0, params[0]]))
    ) < t_params[0]
    return done
```

Figure 12: Skill 0 for open drawer.

```python
def compute_dense_reward_1(self, action, obs, params) -> float:
    params = []
    # Reward term 1: The negative distance between the robot's end-effector and the
    # drawer box handle.
    reward_term_1 = -np.linalg.norm(obs['current_state'][:3] - \
        obs['current_state'][14:17])
    # Reward term 2: Regularization term on the robot's action.
    reward_term_2 = -np.linalg.norm(action)
    # The total reward is the weighted sum of the reward terms.
    reward = 1 * reward_term_1 + 0.1 * reward_term_2
    return reward

def termination_1(self, obs, params, t_params) -> bool:
    t_params = [0.01]
    # The skill is considered successful if the distance between the robot's
    # end-effector and the drawer box handle is less than a threshold.
    done = np.linalg.norm(obs['current_state'][:3] - \
        obs['current_state'][14:17]) < t_params[0]
    return done
```

Figure 13: Skill 1 for open drawer.

```python
def compute_dense_reward_2(self, action, obs, params) -> float:
    params = []
    # Reward term 1: The negative difference between the current state of the drawer
    # box and its goal state.
    reward_term_1 = -np.abs(obs['current_state'][17] - (-0.16))
    # Reward term 2: Regularization term on the robot's action.
    reward_term_2 = -np.linalg.norm(action)
    # The total reward is the weighted sum of the reward terms.
    reward = 1 * reward_term_1 + 0.1 * reward_term_2
    return reward

def termination_2(self, obs, params, t_params) -> bool:
    t_params = []
    # The skill is considered successful if the drawer box is fully open.
    done = obs['current_state'][17] < -0.15
    return done
```

Figure 14: Skill 2 for open drawer.

**Stack cube.** The stack cube task in the ManiSkill2 is one of the most complicated task in our experiments. To solve this task, the LLM provides a chain of skills from skill 0 all the way to skill 4 (see Fig. 15, 16, 17, 18 and 19). The corresponding response from LLM is:

*The skills and sequence of skills for accomplishing each simple task are:*

*- Skill 1: Navigate gripper to cube A.*

*- Skill 2: Grasp cube A.*

*- Skill 3: Lift cube A in the air.*

*- Skill 4: Move cube A to the position above cube B.*

*- Skill 5: Release cube A.*

*The sequence for accomplishing the task could be: Skill 1 -> Skill 2 -> Skill 3 -> Skill 4 -> Skill 5.*

```python
def compute_dense_reward_0(self, action, obs, params) -> float:
    import numpy as np
    # Reward term 1: the distance between robot's gripper and cube A
    dist_to_cubeA = np.linalg.norm(self.tcp.pose.p - self.cubeA.pose.p)
    # Reward term 2: regularization of the robot's action
    action_reg = np.linalg.norm(action)
    reward = -dist_to_cubeA - 0.1 * action_reg
    return reward

def termination_0(self, obs, params, t_params) -> bool:
    t_params = [0.01]
    dist_to_cubeA = np.linalg.norm(self.tcp.pose.p - self.cubeA.pose.p)
    done = dist_to_cubeA < t_params[0]
    return done
```

Figure 15: Skill 0 for stack cube.

```python
def compute_dense_reward_1(self, action, obs, params) -> float:
    import numpy as np
    # Reward term 1: the distance between robot's gripper and cube A
    dist_to_cubeA = np.linalg.norm(self.tcp.pose.p - self.cubeA.pose.p)
    # Reward term 2: regularization of the robot's action
    action_reg = np.linalg.norm(action)
    reward = -dist_to_cubeA - 0.1 * action_reg
    return reward

def termination_1(self, obs, params, t_params) -> bool:
    done = self.agent.check_grasp(self.cubeA)
    return done
```

Figure 16: Skill 1 for stack cube.

```python
def compute_dense_reward_2(self, action, obs, params) -> float:
    import numpy as np
    # Reward term 1: the height of cube A
    height_of_cubeA = self.cubeA.pose.p[2]
    # Reward term 2: regularization of the robot's action
    action_reg = np.linalg.norm(action)
    reward = height_of_cubeA - 0.1 * action_reg
    return reward

def termination_2(self, obs, params, t_params) -> bool:
    t_params = [0.1]
    done = self.cubeA.pose.p[2] > t_params[0]
    return done
```

Figure 17: Skill 2 for stack cube.

```python
def compute_dense_reward_3(self, action, obs, params) -> float:
    import numpy as np
    # Reward term 1: the distance between cube A and the position above cube B
    dist_to_above_cubeB = np.linalg.norm(self.cubeA.pose.p - \
        (self.cubeB.pose.p + np.array([0, 0, 0.02])))
    # Reward term 2: regularization of the robot's action
    action_reg = np.linalg.norm(action)
    reward = -dist_to_above_cubeB - 0.1 * action_reg
    return reward

def termination_3(self, obs, params, t_params) -> bool:
    t_params = [0.01]
    dist_to_above_cubeB = np.linalg.norm(self.cubeA.pose.p - \
        (self.cubeB.pose.p + np.array([0, 0, 0.02])))
    done = dist_to_above_cubeB < t_params[0]
    return done
```

Figure 18: Skill 3 for stack cube.

```python
def compute_dense_reward_4(self, action, obs, params) -> float:
    import numpy as np
    # Reward term 1: the openness of robot gripper
    gripper_openness = self.agent.robot.get_qpos()[-1] / \
    self.agent.robot.get_qlimits()[-1, 1]
    # Reward term 2: if coub A is on cube B
    cubeA_on_cubeB = 1 if self.check_cubeA_on_cubeB() else -1
    # Reward term 3: regularization of the robot's action
    action_reg = np.linalg.norm(action)
    reward = gripper_openness + cubeA_on_cubeB - 0.1 * action_reg
    return reward

def termination_4(self, obs, params, t_params) -> bool:
    done = not self.agent.check_grasp(self.cubeA) and \
        self.check_cubeA_on_cubeB() and check_actor_static(self.cubeA)
    return done
```

Figure 19: Skill 4 for stack cube.

