# OpenReview forum: "Skill Discovery using Language Models"
_ICLR.cc/2025/Conference — Submitted to ICLR 2025_

### Official Review · Reviewer_kCBF · 2024-10-21

**Soundness:** 3
**Presentation:** 2
**Contribution:** 3
**Rating:** 6
**Confidence:** 4

**Summary:**

This paper presents an approach called  Language2Skills that decomposes a natural language task description into reusable skills.  Each skill is defined by an LLM generated dense reward function, which can be used for skill policy training. L2S aims to make it easier for RL practitioners to develop reward and termination functions for training RL agents in multi-step problems.

**Strengths:**

- The proposed method is a novel extension of previous works like T2R and Eureka to the multi-step setting, which was a nontrivial extension due to the dependencies between skills which required an additional optimization step.

- The results show that the proposed method (L2S) outperforms those baselines in many multi-step RL problems.

- The problem formulation and method/algorithm were clearly described and mathematically precise.

**Weaknesses:**

- The abstract was somewhat difficult to understand. Specifically the following sentence: “To address the uncertainty surrounding the parameters used by the LLM agent in the generated reward and termination functions, L2S trains parameter-conditioned skill policies that performs well across a broad spectrum of parameter values.” — What are the parameters being referred to here? Why is there uncertainty over them? I was able to understand the abstract after reading the paper, but it wasn’t a strong introduction to the material.

- I’m not following the author’s critique of existing skill chaining literature: “For example, learning a skill π1 to move an object towards a goal region cannot be learned well before mastering the skill π2 for object grasping, but skill chaining would require learning π1 first”. Why is this the case? Since all learning is done in a simulator, presumably the block could simply be initialized in the hand for training π1 before training π2.

- No error bars in figure 5 and no statistical analysis for any of the results.

- It looks like getting this to work required a considerable amount of task-specific prompt engineering. I’m concerned that such a system is not able to generalize to new tasks without significant additional “tips and tricks” added into the prompt:
“1.) Tasks about moving mug are considered successful when mug is moved at least 0.1 meter towards the correct direction compared with the object's initial position. 2.) Tasks about turning faucet are considered successful when faucet is turned at least np.pi/4 radian towards the correct direction compared with the object initial position. 3.) Tasks about opening or closing drawer are considered successful when drawer box is fully open or fully closes. Drawer fully open means drawer box position is smaller than -0.15 meter. Drawer fully closed means drawer box position is greater than -0.01 meter.“
I’m concerned that the trial-and-error process required to create these sorts of prompts might be more difficult than directly writing the reward and termination functions.

- There is no supplementary code, as encouraged by ICLR, which makes reproducibility difficult to assess.

Grammatical Issues:

- Abstract: "Large Language models (LLMs) possess remarkable ability" should be "Large Language Models (LLMs) possess a remarkable ability" (article "a" is needed before "remarkable ability").

- Abstract: "short of Language2Skills" should be "short for Language2Skills.”

- Page 1: "However, it is known prone to inefficient task decomposition..." should be "However, it is known to be prone to inefficient task decomposition...” known prone -> known to be prone

- Page 2: "L2S adopts a strategy of training a skill policy that performs well across a broad spectrum of parameter values and select the most suitable parameter value during the training of subsequent skills." select -> selects

- Page 2: "turn fauccet left" -> "turn faucet left”

**Questions:**

- In section 3.2 the authors state that they do few-shot prompting with examples, pointing to Appendix E for details. However, I don’t see any examples in Appendix E. Appendix F is titled “Examples of Reward Functions and Termination Condition Functions”, but it seems like those are examples of the LLM output rather than examples used in the prompt?

- How are the “user-selected” variances for the skill parameters determined? Wouldn’t the variances be task and subtask-specific? I would imagine if the variances are too big, the policy may not learn anything and if the variances are too small the skill would not generalize.

- In equation 1, wouldn’t optimizing \phi_{i-1} and \varphi_{i-1} potentially lead those variables to be out of distribution for the frozen policy \pi_{i-1}?

- In the unnumbered equation between equations 1 and 2, how do you handle policies that don’t terminate? It looks like i(\tao, \beta) is undefined in that case. Is there an implicit assumption that all policies will eventually terminate? How is that enforced?

- Why make the assumption that the parameters for each skill only depend on the skill after them? What if there are constraints across non-adjacent skills in the chain? For example, if the fingers needed to be closed to push a button, the sequence may be 1.) move to button, 2.) Close fingers, 3.) Push button. However, the “move to button” parameters could not be changed when training the push button skill.

- Why optimize the parameters and the policy separately in an iterative process rather than together?

Overall, the paper introduces a novel and interesting approach with favorable comparisons to baselines. I would be happy to increase my score if the authors address my questions, include error bars or statistical analysis in their comparisons, improve readability, and provide code for reproducibility.

---

> ### Author Response · Authors · 2024-11-24
> **Reply to Questions part (Part 1)**
>
> Thank you for your dedication to reading our work thoroughly and providing thoughtful feedback. Your insights have been instrumental in helping us refine and improve our research. Below are our responses to the weaknesses and questions you raised.
> We provide our essential code base at https://anonymous.4open.science/r/L2S_basecode-F5D4.
>
> ## Questions
>
> > **Few-shot prompting with examples**
>
> Thank you for pointing out the missing part. We provide the content of our few-shot examples here and will include them in Appendix E. These few-shot examples serve two purposes: 1) to provide workflows for task decomposition, and 2) to constrain the LLM from generating overly fine-grained or meaningless skills that contribute little to the task.
>
> ```markdown
> Instances of Few-shot Examples:
> 1.Task to be fulfilled: Turn an object with a handle left.
> Corresponding skills and sequence of skills for accomplishing the task:
>   Skill 1: Align the robot arm end-effector to a 3D position on the right of the object handle with some offset.
>   Skill 2: Move robot arm end-effector and turn the object handle left.
>   The sequence for accomplishing the task could be: Skill 1 -> Skill 2.
> 2.Task to be fulfilled: In the MuJoCo PickAndPlace environment, pick up a box and move it to the 3D goal position and hold it there.
> Corresponding skills and sequence of skills for accomplishing the task :
>   Skill 1. Navigate gripper to the box.
>   Skill 2. Grasp the box and move the box to the goal position and hold it.
>   The sequence for accomplishing the task could be: Skill 1 -> Skill 2.
> ```
>
> > **Policy Termination**
>
> As is common in reinforcement learning methods, a policy terminates either when the termination condition is met or when the maximum number of action steps is reached. The state distribution after the policy terminates, regardless of the termination reason, is then considered as the initial state distribution for training the next skill.
>
> > **Why make the assumption that the parameters for each skill only depend on the skill after them?**
>
> Repeatedly optimizing the parameters of previously learned skills can be inefficient, as it enlarges the search space. More importantly, optimizing the parameters of all prior skills simultaneously can degrade the performance of some skills, as each skill is trained based on an initial state distribution conditioned on the optimized parameter values of the preceding skills. Updating these parameters for earlier skills may lead to suboptimal behavior.To support this explanation, we conducted additional evaluations using the "Open Drawer" task from LORL and the "Pick Cube" task from ManiSkill2, each of which involves three decomposed skills. In these evaluations, we optimized the parameters of both the first two skills while training the third skill. Results indicate that optimizing all prior parameters incurs higher training costs to maintain a similar success rate:
>
> ***Training of third skill in LORL-Open Drawer***
>
> |                                   | Success Rate | Training Cost on 3rd skill (Timesteps) |
> | --------------------------------- | ------------ | -------------------------------------- |
> | Default                           | 0.95(0.014)  | 4e5                                    |
> | Optimizing all the previous skill | 0.96(0.011)  | 1e6                                    |
>
> ***Training of third skill in ManiSkill2-Pick Cube***
>
> |                                   | Success Rate | Training Cost on 3rd skill (Timesteps) |
> | --------------------------------- | ------------ | -------------------------------------- |
> | Default                           | 0.87(0.04)   | 2e6                                    |
> | Optimizing all the previous skill | 0.813(0.01)  | 3e6                                    |
>
> Given a subtask sequence of 1) move to button, 2) close fingers, and 3) push button, we assume that the reward and termination functions for skill 2 developed by the LLM encourage and require the end effector to be fully closed around the button.
>
> > **Why optimize the parameters and the policy separately in an iterative process rather than together?**
>
> When the parameters and policy are optimized together, changes to the policy might invalidate the parameters, leading to potential suboptimal behavior. Optimizing them separately allows for the preservation of previously learned behaviors while still refining the system.

---

> ### Author Response · Authors · 2024-11-24
> **Reply to Questions part (Part 2)**
>
> > **In equation 1, wouldn’t optimizing \phi_{i-1} and \varphi_{i-1} potentially lead those variables to be out of distribution for the frozen policy \pi_{i-1}?**
>
> When training the $i$-th skill $\pi_i$, optimizing the parameters $\phi_{i-1}$ and $\varphi_{i-1}$ from the previous $(i-1)$-th skill does not lead to initial states outside the distribution of the frozen policy $\pi_{i-1}$. In L2S, the skill policies are parameter-conditioned, meaning that, during training, they are trained to handle a range of parameter values well (similar to goal-conditioned RL). When optimizing $\pi_i$, these parameter values $\phi_{i-1}$ and $\varphi_{i-1}$ from $\pi_{i-1}$ are adjusted within the same range, maintaining stability during training.
>
> > **How are the “user-selected” variances for the skill parameters determined?**
>
> As we provided information about the environment and additional knowledge that connects the semantics of real-world instructions to the robot environment and specifies the task's successful conditions (see Appendix E), the LLM gains some understanding of the environment's scale and selects reasonable (though not necessarily optimal) parameter mean values. By default, we set the variance of the parameters to be twice the maximum mean value generated by the LLM for the current task (a heuristic). We conducted experiments with varying alternative parameter value variances, while keeping the parameter mean value fixed:
>
> ***LORL-Turn faucet left***
>
> For this task, we examined three different parameter variance combinations—variants 1, 2, and 3—to analyze their effects on reward and termination parameters. The first skill in the task is trained to position the end effector around the faucet handle, with the reward function parameter defining the acceptable distance to the handle. The termination condition parameter specifies how close the end effector must be to the target position to transition to the next skill.
>
> - **Variant 1**: The variance of the termination condition parameter is increased.
> - **Variant 2**: The variance of the reward function parameter is increased.
> - **Variant 3**: The variance of both parameters is increased.
>
> | [Reward Func Parameters]/[Termination Func Parameters] | Initial Parameters Mean Value | Initial Parameters Variance | Optimized Params/(Std)         | Success Rate/(std) | Training Cost (Timesteps) |
> | ------------------------------------------------------ | ----------------------------- | --------------------------- | ------------------------------ | ------------------ | ------------------------- |
> | Default                                                | [0.01]/[0.01]                 | [0.2]/[0.02]                | [0.107/(0.02)]/[0.013/(0.001)] | 0.99/(0.01)        | 1e6                       |
> | **Variant1**                                           | [0.01]/[0.01]                 | [0.2]/**[0.05]**            | [0.118/(0.04)]/[0.027/(0.005)] | 0.89/(0.13)        | 1.5e6                     |
> | **Variant2**                                           | [0.01]/[0.01]                 | **[0.5]**/[0.02]            | [0.16/(0.01)]/[0.015/(0.0003)] | 0.97/(0.04)        | 1e6                       |
> | **Variant3**                                           | [0.01]/[0.01]                 | **[0.5]/[0.05]**            | [0.114/(0.01)]/[0.031/(0.001)] | 0.93/(0.05)        | 1e6                       |
>
> The results were obtained using three different random seeds and demonstrate that L2S consistently achieves optimal parameter values across the default setting and all tested variants.
>
> ***ManiSkill2-Open drawer***
>
> For this task, the parameter in the termination condition of the first skill specifies the required proximity of the robot's end effector to the target position above the drawer handle. In the variant, we increase the variance of this parameter from the default value of 0.02 to 0.05 to evaluate its impact on performance.
>
> | [Termination Func] | Initial Params Mean Value | Initial Params Variance | Optimized Params(std) | Success Rate/(Std) | Training Cost (Timesteps) |
> | ------------------ | ------------------------- | ----------------------- | --------------------- | ------------------ | ------------------------- |
> | Default            | [0.01]                    | [0.02]                  | [0.026(0.003)]        | 0.94/(0.06)        | 1e6                       |
> | **Variant**        | [0.01]                    | **[0.05]**              | [0.031(0.006)]        | 0.97/(0.02)        | 1e6                       |
>
> The results, obtained using three different random seeds, consistently demonstrate the robustness of L2S in handling various parameter variance configurations while maintaining effective performance.

---

> ### Author Response · Authors · 2024-11-24
> **Reply to Weaknesses part (Part 3)**
>
> > **The abstract was somewhat difficult to understand.**
>
>  We will update the paper to reflect the role of parameters in policies in the abstract.
>
> > **The critique of existing skill chaining literature**
>
> We will clarify in the paper that we do not assume the system can be reset to any state, as this would be a strong assumption for the learning algorithm in the real-world setting.
>
> > **No error bars in figure 5 and no statistical analysis for any of the results.**
>
> Thanks for pointing out the inadequacy of result analysis! We will add error bars in Fig. 5 and update the statistical analysis for Fig. 4 and Fig. 5 in the article as follows:
>
> “For the performance on the task sequence of 6 tasks, as shown in Fig. 4: 1) In LORL, L2S solved an average of 5.44 tasks in a total of 1.1e7 time steps, while L2S-fixed solved 4.98 tasks, and Text2Reward solved 4.9 tasks in a total of 1.5e7 time steps. 2) In ManiSkill2, L2S solved an average of 5.33 tasks in a total of 1e7 time steps, while L2S-fixed solved 4.14 tasks, and Text2Reward solved only 2.68 tasks in a total of 1.5e7 time steps. Overall, L2S showed an improvement of 11.0% in LORL and 98.8% in ManiSkill2, while requiring 26.7% and 33.3% less training cost compared to the baseline Text2Reward.For the performance on single simple or complex tasks, as shown in Fig. 5, L2S outperformed the baseline Text2Reward by 18.7% and 98.7% on average success rate in LORL and ManiSkill2 environments, respectively, demonstrating a significant performance improvement with L2S.”
>
> >**I'm concerned that the trial-and-error process required to create task-specific prompts might be more difficult than directly writing the reward and termination functions.**
>
> The meaning and intention of this part of the prompt seem to have been misunderstood. The user-provided “additional environment knowledge” is meant to bridge the semantics of real-world instructions to the robotics environment and define the task's overall success conditions. Specifically, it involves: 1) mapping the real-world directions (e.g., Cartesian coordinate axes) to the task description, and 2) specifying the success criteria for the task, which can vary based on human preferences. For example, in the LORL environment, the human instruction "moving an object forward" should correspond to moving the object along the positive y-axis in the simulation, and the task would be considered successful if the object is moved 0.1 meters along the positive y-axis.

---

> ### Comment · Reviewer_kCBF · 2024-11-27
> **Reply**
>
> Thanks to the authors for answering my questions, providing the codebase, and including error bars in their experimental results, and including the missing appendix section.
>
> Regarding the assumption that each skill only depend on the skill after them, it is clear from the experiments that the assumption holds for the tasks that were evaluated on. However, I don't think this assumption holds in a lot of other tasks. The fact that "optimizing the parameters of all prior skills simultaneously can degrade the performance of some skills" is a limitation of this particular method on tasks where the assumption doesn't hold. This is maybe a direction for future work, but should probably be listed as a limitation.
>
> > The user-provided “additional environment knowledge” is meant to bridge the semantics of real-world instructions to the robotics environment and define the task's overall success conditions
>
> This is reasonable for some of the "additional environment knowledge" and not as reasonable for others. For example "3. Tasks about moving mug are considered successful when mug is moved at least 0.1 meter towards the correct direction compared with the object's initial position." This is an extremely task-specific piece of context. I think the extensive prompt engineering required to get this system to work should be listed as one of the limitations.
>
> Lastly, the abstract remains unchanged and continues to contain somewhat vague terminology such as "uncertainty surrounding the parameters used by the LLM agent" that is hard to understand without first understanding the paper.
>
> Overall, I think the paper has improved and I have increased my score accordingly.

---

> ### Author Response · Authors · 2024-11-29
>
> Thank you once again for your thoughtful response, suggestions, and the score increase! We have included “optimizing the required previous skills (not just the immediate one)” as a limitation and potential extension in the paper.
>
> Regarding the concern that some additional knowledge appears highly task-specific—such as the example, *“Tasks involving moving a mug are considered successful when the mug is moved at least 0.1 meters in the correct direction compared to its initial position”*—this type of success condition can be provided as input alongside the task description. That said, Algorithm 1 (line 7-9) in the paper can utilize sparse reward feedback from the environment to automatically infer such termination conditions, even if they are not explicitly specified in the prompt. We hope this explanation helps mitigate the concern.
>
> We will update the abstract of our paper as follows:
>
> "... Each skill is defined by an LLM-generated dense reward function and a termination condition, which in turn lead to effective skill policy training and chaining for task execution. **However, LLMs lack detailed insight into the specific low-level control intricacies of the environment, such as threshold parameters within the generated reward and termination functions. To address this uncertainty,**  L2S trains parameter-conditioned skill policies that performs well across a broad spectrum of parameter values. ..."

---

### Official Review · Reviewer_1Mn6 · 2024-10-23

**Soundness:** 2
**Presentation:** 3
**Contribution:** 1
**Rating:** 6
**Confidence:** 5

**Summary:**

This paper proposes L2S (Language to Skills) to leverage LLMs to decompose tasks into reusable skills with parameterized rewards and termination conditions​. With parameterized rewards and termination conditions, L2S can train parameter-conditioned skill policies and then autonomously accumulates a skill library for guiding continual skills learning.

However, the high-level idea of L2S is almost the same as a previous ICLR LLMAgent Workshop paper [1]. It limits the contribution of this paper. Also, this paper even didn't cite the previous paper. **The author needs to give sufficient explanation to show that it's just inadequate reference or any more serious problem.**

[1]. Li et al., LEAGUE++: EMPOWERING CONTINUAL ROBOT LEARNING THROUGH GUIDED SKILL ACQUISITION WITH LARGE LANGUAGE MODELS, 2024

**Strengths:**

1. Skill reuse between tasks can enhance the training efficiency for continual learning on new tasks.

2. Decompose tasks into smaller reusable skills reduce the difficulties to generate multi-stage dense rewards and enhance training efficiency.

3. The experimental results show that L2S not only solves continuously presented tasks much faster but also achieves higher success rates compared to state-of-the-art methods.

**Weaknesses:**

1. The high-level idea of this paper is almost the same as a previous ICLR Workshop paper [1]. League++ integrates LLMs, Task and Motion Planning (TAMP), and RL to guide continual skill acquisition through task decomposition, reward generation, and continual learning with skills library. Then, League++ can do autonomous training for long-horizon task and enhance training efficiency in other tasks. The overall ideas and contributions are very similar to L2S. However, This paper even didn't cite League++. **The author needs to give sufficient explanation to show that it's just inadequate reference or any more serious problem.** Anyway, the author needs to highlight the difference and show the novelty of this paper. On the other hand, Gen2Sim [2] is also utilizes LLMs to do task generation, task decomposition, and reward generations for skill learning. Please add more related work.

2. In the previous work Text2Reward [3], the reward generation requires human-in-loop training for generating better rewards. Although L2S uses methods similar to Text2Reward, this paper didn't show anything about it.

3. In the previous work Text2Reward [3], the generated reward code can be incorrect. As shown in the table 7 of this previous paper, the success rate is just 90%. Since L2S will generate several reward functions for different skills, the failure rates will exponential growth. (For example, if you have five skills, the success rate will be 90%^5=59%, which is pretty low) This paper didn't show how do they resolve this problem. Adding more explanations and ablation studies is helpful.

4. It's not convincing that only optimize the last skill parameters is enough. (For example, you have a task with three skills: open the cabinet, pick a hammer, and place the hammer to the cabinet. If the cabinet is not opened far enough, you cannot place the hammer successfully. At the moment, you need to update the parameters and termination of the first skill, not the last skill.) Some previous work refine all the skills start from the beginning.

5. The most long-horizon task in the experiments only has three stages: OpenDrawer, PlaceCubeDrawer and CloseDrawer. Since the idea for task decomposition and skill library are designed for long-horizon tasks, more experiments on them should benefit.

[1]. Li et al., LEAGUE++: EMPOWERING CONTINUAL ROBOT LEARNING THROUGH GUIDED SKILL ACQUISITION WITH LARGE LANGUAGE MODELS, 2024

[2]. Katara et al., Gen2Sim: Scaling up Robot Learning in Simulation with Generative Models, 2024

[3]. Xie et al., TEXT2REWARD: REWARD SHAPING WITH LANGUAGE MODELS FOR REINFORCEMENT LEARNING, 2024

**Questions:**

1. In the previous work Text2Reward [1], the reward generation requires human-in-loop training for generating better rewards. It's hard to generate usable reward functions in one-shot. Although L2S uses methods similar to Text2Reward, this paper didn't show anything about it. How can you generate usable reward codes in one shot?

2. The comparison with Eureka [2] is confused for me. Different from T2R, Eureka is a two-stage method, which learns a dense rewards at first and then use it for RL training. How do you compare the training efficiency directly? Also, why the performance of a single skill like OpenDrawer is pretty low? (It shows great performance in much more complex tasks.)

3. The potential assumption of the skill library is that the similar skills' semantic description are similar. It's might be a correct assumption that skill with similar semantic description can be reused. However, it might not be sufficient. How about those skills have similar motions? (like place and insert might have similar motion somehow) How can we utilize them as reuse skills?

If the author can address those limitations and give enough explanation to the novelty, I will consider improve my score.

[1]. Xie et al., TEXT2REWARD: REWARD SHAPING WITH LANGUAGE MODELS FOR REINFORCEMENT LEARNING, 2024

[2]. Ma et al., Eureka: Human-Level Reward Design via Coding Large Language Models, 2024

**Details Of Ethics Concerns:**

.

---

> ### Author Response · Authors · 2024-11-24
> **Reply to Questions part (Part 1)**
>
> We sincerely appreciate the time and effort you have dedicated to carefully reading our paper, and we sincerely thank you for your valuable feedback. Below are our responses to the weaknesses and questions you raised.
>
> ## Questions
>
> > **How can you generate usable reward codes in one shot?**
>
> L2S generates usable reward functions in one shot by leveraging task decomposition into subtasks (skills) through the LLM. Each subtask requires simpler reward functions, making them easier to define accurately compared to the more complex reward functions needed in Text2Reward, which must handle entire tasks and are more challenging to get right.
>
> > **How do you compare the training efficiency with Eureka directly? Why the performance of a single skill like OpenDrawer is pretty low in Eureka? (It shows great performance in much more complex tasks.)**
>
> Eureka incorporates a feedback loop where successful RL reward function samples from the previous step are used as input to generate new samples from the LLM. For our benchmarks, we ran Eureka for 3 rounds with 8 samples per round. This process resulted in significantly higher training costs compared to L2S, measured by the environment steps required for agent training. While increasing the number of rounds or samples per round could potentially improve Eureka's performance on our benchmarks, we observed that L2S remains more sample-efficient overall. For Eureka, we conducted multiple runs and reported results only from those that had at least one successful sample in each round. Eureka achieves strong performance in the ManiSkill2 OpenDrawer environment but struggles in the LORL OpenDrawer environment. In contrast, L2S performs consistently well in both environments, demonstrating its robustness across a diverse range of tasks and settings.
>
> > **How can we manage skills with distinct semantic descriptions but similar motion patterns (e.g., "place" and "insert")? Additionally, how can we effectively utilize such skills for reuse across tasks?**
>
> L2S decomposes complex tasks into subtasks, enabling skill reuse. For instance, it may break down tasks like "place" and "insert" into common subtasks such as "moving closer to the object," "grasping the object," and "moving closer to the target." The final steps, however—such as "putting it on the target position" for "place" and "putting it into the target position" for "insert"—are task-specific and less reusable. Nonetheless, L2S can effectively reuse the shared first 3 subtasks across different tasks, demonstrating its strength in leveraging compositional structures.

---

> ### Author Response · Authors · 2024-11-24
> **Reply to Weaknesses part (Part 2)**
>
> ## Weaknesses
>
> > **Comparison with League++**
>
> Thank you for highlighting the related work we missed. In League++, the reward functions are generated by the LLM through selecting and weighting pre-defined metric functions provided by human experts. This approach simplifies the problem and minimizes irrelevant outputs by constraining the solution space. However, L2S differs from League++ in the following key aspects:
>
> 1. **Free-form Reward Generation:** Unlike League++, L2S enables the LLM to generate free-form reward functions directly from the environment knowledge, rather than being limited to a predefined set of metric functions.
>
> 2. **Reduced Human Expert Effort:** L2S requires less human effort, as it only needs natural language examples of task decomposition. Providing these examples is significantly easier than designing specialized metric functions.We will include a discussion of League++ in the related work section of our paper to address this comparison.
>
> > **As shown in Table 7 of Text2Reward, the success rate of generating executable reward functions is just 90%. Since L2S generates several reward functions for different skills, the failure rates grow exponentially.**
>
> L2S achieves a higher execution success rate for each generated skill compared to the whole-task reward function generated in Text2Reward. This is because generating free-form function code for individual skills is inherently simpler than generating a single function for the entire task. Each skill represents only a portion of the overall task, reducing complexity. To substantiate this claim, we evaluated the LLM's performance in generating correct function code for both LORL and ManiSkill2 environments more than 100 samples each:
>
> | LLM(GPT-4)         | LORL | ManiSkill2 |
> | ------------------ | ---- | ---------- |
> | Correct            | 92%  | 87%        |
> | Syntax/Shape Error | 8%   | 13%        |
>
> The results highlight the effectiveness of L2S in breaking down complex tasks into manageable components and improving the reliability of code generation.
>
> > **It's not convincing that only optimize the last skill parameters is enough.**
>
> Repeatedly optimizing the parameters of previously learned skills can be inefficient, as it enlarges the search space. More importantly, optimizing the parameters of all prior skills simultaneously can degrade the performance of some skills, as each skill is trained based on an initial state distribution conditioned on the optimized parameter values of the preceding skills. Updating these parameters for earlier skills may lead to suboptimal behavior. To support this explanation, we conducted additional evaluations using the "Open Drawer" task from LORL and the "Pick Cube" task from ManiSkill2, each of which involves three decomposed skills. In these evaluations, we optimized the parameters of both the first two skills while training the third skill. Results indicate that optimizing all prior parameters incurs higher training costs to maintain a similar success rate:
>
> ***Training of third skill in LORL-Open Drawer***
>
> |                                   | Success Rate | Training Cost on 3rd skill (Timesteps) |
> | --------------------------------- | ------------ | --------------------------------------- |
> | Default                           | 0.95(0.014)  | 4e5                                     |
> | Optimizing all the previous skill | 0.96(0.011)  | 1e6                                     |
>
> ***Training of third skill in ManiSkill2-Pick Cube***
>
> |                                   | Success Rate | Training Cost on 3rd skill (Timesteps) |
> | --------------------------------- | ------------ | --------------------------------------- |
> | Default                           | 0.87(0.04)   | 2e6                                     |
> | Optimizing all the previous skill | 0.813(0.01)  | 3e6                                     |
>
> > **More experiments for long-horizon tasks**.
>
> To address the reviewer’s concern, we provided additional results on complex, meaningful tasks in both the LORL and ManiSkill2 benchmarks. Notably, we prompted GPT-4 in both L2S and Text2Reward to reuse policies learned from prior single tasks whenever possible, ensuring a fair comparison between the two approaches.
>
> | Benchmark  | Task                                             |  Text2Reward(Std)   | L2S(Std) |
> | ---------- | ------------------------------------------------ | ----------- | ---------------- |
> | LORL       | PushMugBack-OpenDrawer                           | 0.91(0.043) |    0.93(0.030)   |
> |            | OpenDrawer-TurnFaucetRight                       | 0.89(0.096) |   0.93(0.062)    |
> |            | MugBack-OpenDrawer-TurnFaucetRight               | 0.76(0.071) |     0.90(0.044)  |
> | ManiSkill2 | OpenDrawer-Place**TwoCubes**Drawer-CloseDrawer   | 0.01(0.002) |    0.72(0.056)   |
> |            | OpenDrawer-Place**ThreeCubes**Drawer-CloseDrawer | 0.01(0.001) |   0.54(0.032)    |

---

> ### Comment · Reviewer_1Mn6 · 2024-11-24
>
> Thanks for the response. With more explanations and experiments, the current version should be better now. I would like to consider increasing my score.
>
> For this version, I still have some questions and suggestions:
>
> 1. For Text2Reward[1], based on my personal experience, the one-shot reward generations are usually even not usable (the generated values and components are unresonable without human-in-loop prompting.) I understand that subtask requires simpler reward functions, making them easier to define accurately, but it would be helpful if you can provide more analysis, examples, or even ablation studies between them.
>
> 2. I think adding more details for implenting Eureka[2] for fair comparison in the draft is important, since it has different training progress rather than L2S and T2R[1].
>
> 3. I think adding more comparsion/analysis between L2S and League++[3] is helpful. I might be curious about the following aspects:
>
>       a. League++[3] combines symbol operators with LLM for skills decompositions and terminal conditions generation, which might be able to increasing the planning success rates but requires more human efforts. I think comparing it with the method used in L2S is also needed. Add some experiments to test the success rates of skill decomposition with L2S is also helpful.
>
>       b. Adding more analysis about the trade-off with higher success rate in generating rewards with human-defined metrics functions (maybe helpful for long-horizon task) or with free-form reward generation with lower success rate in generating reward is helpful. Also, in League++[3], actually  the "skills" shown in the L2S are similar to the "metrics functions". Adding some analysis regarding this choice is helpful.
>
>       c. Since the ideas have a lot of similarity, change the story and highlight the contribution will enhance the novelty.
>
> 4. For only optimizing the last skill, it's impressive to see that optimizing all prior parameters incurs higher training costs to maintain a similar success rate. However, with my example shown before (you have a task with three skills: open the cabinet, pick a hammer, and place the hammer to the cabinet. If the cabinet is not opened far enough, you cannot place the hammer successfully. At the moment, you require to update the parameters and termination of the first skill, not the last skill), I think it's necessary to figure out which skill required to be optimized. To let the framework figure out which skills need to be optimized might be a great extension
>
> 5. I am not quite understand the details about the reward generation ablation. For reward generation experiment in LORL and ManiSkill2 environments, what are the tasks you use? How many stage/skills do they have? The success rate is for single skills or the entire tasks? Also, looks like the shape/syntax errors in T2R is only 3%. How about the other errors shown in T2R?
>
> [1]. Xie et al., TEXT2REWARD: REWARD SHAPING WITH LANGUAGE MODELS FOR REINFORCEMENT LEARNING, 2024
>
> [2]. Ma et al., Eureka: Human-Level Reward Design via Coding Large Language Models, 2024
>
> [3]. Li et al., LEAGUE++: EMPOWERING CONTINUAL ROBOT LEARNING THROUGH GUIDED SKILL ACQUISITION WITH LARGE LANGUAGE MODELS, 2024

---

> > ### Author Response · Authors · 2024-11-26
> > **Reply to questions and suggestions (Part 1)**
> >
> > > **For Text2Reward, based on my personal experience, the one-shot reward generations are usually even not usable**
> >
> > Text2Reward can fail when combining reward components from different steps, which is avoided in our method. For example, in the ManiSkill2 "PickCube" task, Text2Reward may sample the following reward function:
> >
> > **"**
> >
> > **r1 = distance_gripper_cube**
> >
> > **r2 = gripper_grasp_cube**
> >
> > **r3 = distance_gripper_pickgoal**
> >
> > **reward = - w1 * r1 + w2 * r2 - w3 * r3**
> >
> > **"**
> >
> > where w1, w2, and w3 are positive values.
> >
> > While each individual reward term might seem reasonable, combining them in this way can make the overall reward function unusable. L2S avoids this issue by ensuring that reward terms from different steps are not mixed together. Instead, they are trained and composed as individual skills, which prevents potential conflicts and maintains the integrity of each skill's reward function.
> >
> > >  **Adding more details for implementing Eureka for fair comparison in the draft is important**
> >
> > We updated the paper to include additional details about implementing Eureka for a fair comparison, as explained in our response above.
> >
> > > **Adding more comparison/analysis between L2S and League++[3] is helpful.**
> >
> > We could not locate an open-source implementation of League++, so we implemented it based on our understanding to conduct additional experiments for comparison.
> >
> > - During the implementation, we limited the LLM to selecting from a set of Metric Functions similar to those explicitly listed in the League++ paper. Here are the Metric Functions we designed for this comparison:
> >
> >   | **Metrics Function**       | **Definition**                                            |
> >   | -------------------------- | --------------------------------------------------------- |
> >   | *dis_to_obj*               | Distance between the gripper and the object               |
> >   | *perpendicular_dis*        | Distance between object and gripper along the normal line |
> >   | *in_grasp*                 | If the object is grasped by the gripper                   |
> >   | drawer_opened (LORL)       | The drawer is opened enough or not.                       |
> >   | faucet_turned (LORL)       | The faucet handle is turned away enough or not.           |
> >   | mug_pushed (LORL)          | The mug is pushed away enough or not.                     |
> >   | *cube_placed* (ManiSkill2) | The cube is placed in the drawer or not.                  |
> >
> >  - Due to the lack of open-sourced code, we were unable to implement the feedback loop described in League++. Instead, we manually selected high-quality LLM-generated skill samples for RL training, which might not fully replicate the intended process in League++.
> >
> > Our evaluation includes the 1) "Open Drawer," 2) "Turn Faucet Left," and 3) "Push Mug Back" tasks from the LORL benchmark, as well as the "Place Cube Drawer" task from the ManiSkill2 benchmark. Skill policies are learned using League++ generated reward functions, together with parameter optimizing method from L2S. All results were obtained using three different random seeds to ensure reliability. League++ underperforms compared to L2S in our implementation:
> >
> > | **Tasks**                    | **League++ Success Rate(Std)** | **L2S Success Rate(Std)** |
> > | ---------------------------- | ------------------------------ | ------------------------- |
> > | LORL-Open Drawer             | 0.83(0.06)                     | 0.95(0.02)                |
> > | LORL-Turn Faucet Left        | 0.58(0.15)                     | 0.99(0.01)                |
> > | LORL-Push Mug Back           | 0.13(0.13)                     | 0.96(0.04)                |
> > | ManiSkill2-Place Cube Drawer | 0.78(0.07)                     | 0.92(0.07)                |
> >
> > League++'s reliance on predefined Metric Functions may limit its ability to capture the full complexity of tasks or appropriately penalize suboptimal behavior, potentially reducing overall performance. This result reinforces the advantage of using free-form reward functions to improve the effectiveness and adaptability of skill training.
> >
> > > **It's necessary to figure out which skill required to be optimized.**
> >
> > Thank you for the insightful suggestion! Algorithm 1 in L2S can indeed be extended to support this approach. If optimizing the parameters for the immediately preceding skill does not yield a satisfactory success rate, the algorithm can iteratively optimize the parameters of earlier skills. These earlier skills are selected based on their interaction with a subset of the objects (as identified from environment information) involved in the current skill.

---

> > ### Author Response · Authors · 2024-11-26
> > **Reply to questions and suggestions (Part 2)**
> >
> > >  **I am not quite understand the details about the reward generation ablation.**
> >
> > For the reward generation experiment in the LORL and ManiSkill2 environments, we selected 5 simple tasks from each environment(LORL: "OpenDrawer, TurnFaucetLeft, TurnFaucetRight, PushMugBack, PushMugLeft"; ManiSkill2: "OpenDrawer, CloseDrawer, PickCube, StackCube, PlaceCubeDrawer"), as shown in Figure 5 in the paper, and queried the LLM for 20 samples per task. Across these 10 tasks, the number of skills generated ranged from 2 to 5. The reported results reflect the success rate for completing the entire tasks.
> >
> > Generating free-form function code for individual skills proved to be inherently simpler than generating a single function for the entire task. As a result, within individual skill functions, we did not encounter errors such as "Wrong package," "Attribute hallucination," or "Class attribute misuse," which were observed in Text2Reward. The syntax errors we experienced were primarily due to the increased difficulty for the LLM in constructing a sequence of function codes that adhered to the required specific format.

---

> ### Comment · Reviewer_1Mn6 · 2024-11-26
>
> Thanks for giving more detailed explanations and more experimental results. Although I would like to see more comparsion in reducing LLMs Hallucinations with limited symbolic operaters & metrics functions and using freeform generations for planning and long-horizon tasks, I understand it might not be the main contents of this paper. I don't have any other questions now and would like to increase my score.

---

> > ### Author Response · Authors · 2024-11-27
> >
> > Thank you again for your thoughtful response and willingness to consider increasing the score for our paper. We greatly appreciate your insights, particularly regarding the potential of domain-specific knowledge and metric functions to constrain the solution space and reduce LLM hallucinations. We will incorporate your advice into future work and update our paper to include a discussion on this topic. It would be greatly appreciated if the score in the main review could be updated to reflect your current assessment.

---

> > > ### Author Response · Authors · 2024-11-29
> > > **Score Update**
> > >
> > > Dear Reviewer 1Mn6,
> > >
> > > We have included this in the limitation section of the paper:
> > >
> > > *"Lastly, LLM hallucinations present challenges in generating robust free-form reward and termination function code. Constraining code generation within a structured representation, possibly defined by a domain-specific language, might offer a balance between generation stability and the exploration of the reward space."*
> > >
> > > We will also include a complete comparison with League++ extending the response above into the paper (or the appendix, if space is constrained):
> > > 1. **We will design a more diverse set of metric functions** to show how much manual design effort is needed to match the success rate of the free-form reward generation in L2S.
> > > 2. **We will compare the reward function executability** between League++ and L2S, where League++ yields a better success rate due to the introduction of symbolic operators that reduce the reward search space to some extent. This will help highlight the trade-offs between manual design and generative flexibility.
> > >
> > > Thank you so much for your comment that you would like to increase the score for our paper. Please let us know if you are satisfied with these proposed revisions and could update the score in your main review to more explicitly reflect your current assessment of the paper. We appreciate your review, comments, suggestions, and help!

---

> ### Comment · Reviewer_1Mn6 · 2024-11-29
>
> Thanks for the adding more details and your reminding. Since the current version has addressed my questions and added some related works and experiments, I would like to increase my score to boardline accept now. Please remember to include the complete comparison into the paper accordingly

---

### Official Review · Reviewer_jBGA · 2024-11-03

**Soundness:** 3
**Presentation:** 3
**Contribution:** 3
**Rating:** 6
**Confidence:** 2

**Summary:**

The paper introduces **L2S (Language to Skills)**, a novel framework that utilizes large language models (LLMs) for skill discovery and reuse in robotics. Unlike prior approaches requiring predefined skill sets, L2S enables robots to autonomously generate skills from task descriptions, defined by LLM-generated dense reward functions and termination conditions. Each skill is parameter-conditioned to adapt across a range of conditions, with suitable parameters chosen during training of subsequent skills. This adaptive approach mitigates the challenge of incorrect parameter settings.

L2S builds a skill library progressively as it encounters new tasks, allowing it to efficiently tackle novel tasks by reusing previously learned skills. Experimentally, L2S demonstrates superior performance on sequential robotic manipulation tasks compared to baseline methods, achieving faster convergence and higher success rates in complex, multi-step tasks in *LORL-Meta-World* and *ManiSkill2* environments.

**Strengths:**

- The proposed framework, which involves training a set of primitive skills and planning on top of them, is novel.
- The performance improvements over previous state-of-the-art methods, such as T2R(Xie et al., 2023) and Eureka(Ma et al., 2023), are significant, particularly for tasks in the ManiSkill2 environment.
- The reusability of learned skills facilitates faster learning for downstream tasks, as demonstrated notably in the PickCube and StackCube tasks. The explanation behind these results is clear and intuitive.

**Weaknesses:**

- **Questions about the Parameters**

    Based on my understanding, the parameters for each skill play a crucial role in both the reusability of learned skills and in chaining these skills to solve tasks. However, certain aspects remain unclear, raising concerns about the generalizability of the proposed framework. I have the following questions:

    - **Who determines the semantic meaning of the parameters?**

        In the *“turn faucet left”* task, one possible semantic meaning of a parameter could be the distance to the handle. My question is: who chooses this dimension? Is it determined by a human or by the language model (LLM)? If decided by the LLM, what happens if the LLM produces unhelpful dimensions? Do you have any mechanism to prevent this from happening? For instance, the semantic dimension could be something less-related for solving the task, like the rotation of the handle or its distance from the ground. In such cases, trying out different parameters for parameter-conditioned policy might not accelerate training but could instead decrease sample efficiency. Additionally, could you clarify how the total number of parameters for each skill is determined and how this might impacts the training speed of L2S? I think including these details in the paper would be beneficial for better understanding.

    - **How is the parameter range chosen for the experimental results in Section 4?**

        According to the paper, the parameter distributions must be specified by the user. However, it seems challenging to determine the optimal mean and variance for training skills efficiently, especially since users need to specify these values prior to training. How can a user accurately define the parameter range before training begins?

        I understand that L2S can select the best values from the provided distribution during training using Equation (2) in line 305, even if the distribution is not perfectly chosen. However, I believe that choosing the right distribution would significantly reduce the sample needed to reach a certain performance level during training. Since the experiments section lacks detailed information on how the specific distributions \( q_{\phi_k} \), even though this choice greatly impacts training speed, I wonder how these distributions were chosen. I have a concern that if the distribution is chosen manually to have high mean and low variance distrubition around the ground-truth(or best performing) value, we have a possibility that increased training speed partly comes from this.

- **Minor Corrections**
    - *Line 230-232:* The term *“processes”* is somewhat vague and colloquial, making it difficult to understand precisely what it refers to.
    - *Line 374:* Instead of *“converged,”* it might sound more natural to say *“the task is considered solved”* or *“the policy is considered converged.”*
    - *Figure 5:* Adding standard deviation would be helpful.

**Questions:**

Questions are provided in the weaknesses section. Please address these questions if possible!"

---

> ### Author Response · Authors · 2024-11-25
> **Reply to Weaknesses (Part 1)**
>
> We sincerely thank you for your valuable feedback and thoughtful comments! We truly appreciate the time and effort you have dedicated to reviewing our submission. Below, we provide detailed responses to your questions.
> We have updated the flaws mentioned in Minor Corrections in paper and will upload it shortly.
>
> ## Weaknesses
>
> >  **Who determines the semantic meaning of the parameters?**
>
> L2S leverages LLMs to decompose a task into subtasks and generates reward and termination functions for each subtask. We prompt the LLM that when it lacks high confidence in determining a task-specific numerical threshold, it introduces a parameter to represent the uncertainty. For instance, in the task "turn faucet left," one subtask involves positioning the end effector near the faucet handle. The threshold for how close the end effector should be to the handle is task-specific, so the LLM introduces a parameter to define this distance. These parameters are *tied to specific concepts within the reward and termination functions generated by the LLM*. Consequently, optimizing these parameters directly enhances the overall reward performance, as it aligns with the structure and objectives defined by the generated reward functions.

---

> ### Author Response · Authors · 2024-11-25
> **Reply to Weaknesses (Part 2)**
>
> >  **How is the parameter range chosen for the experimental results in Section 4?**
>
> As we provided information about the environment and additional knowledge that connects the semantics of real-world instructions to the robot environment and specifies the task's successful conditions (see Appendix E), the LLM gains some understanding of the environment's scale and selects reasonable (though not necessarily optimal) parameter mean values. By default, we set the variance of the parameters to be twice the maximum mean value generated by the LLM for the current task (a heuristic). We conducted experiments with varying alternative parameter value variances, while keeping the parameter mean value fixed:
>
> ***LORL-Turn faucet left***
>
> For this task, we examined three different parameter variance combinations—variants 1, 2, and 3—to analyze their effects on reward and termination parameters. The first skill in the task is trained to position the end effector around the faucet handle, with the reward function parameter defining the acceptable distance to the handle. The termination condition parameter specifies how close the end effector must be to the target position to transition to the next skill.
>
> - **Variant 1**: The variance of the termination condition parameter is increased.
> - **Variant 2**: The variance of the reward function parameter is increased.
> - **Variant 3**: The variance of both parameters is increased.
>
> | [Reward Func Parameters]/[Termination Func Parameters] | Initial Parameters Mean Value | Initial Parameters Variance | Optimized Params/(Std)         | Success Rate/(std) | Training Cost (Timesteps) |
> | ------------------------------------------------------ | ----------------------------- | --------------------------- | ------------------------------ | ------------------ | ------------------------- |
> | Default                                                | [0.01]/[0.01]                 | [0.2]/[0.02]                | [0.107/(0.02)]/[0.013/(0.001)] | 0.99/(0.01)        | 1e6                       |
> | **Variant1**                                           | [0.01]/[0.01]                 | [0.2]/**[0.05]**            | [0.118/(0.04)]/[0.027/(0.005)] | 0.89/(0.13)        | 1.5e6                     |
> | **Variant2**                                           | [0.01]/[0.01]                 | **[0.5]**/[0.02]            | [0.16/(0.01)]/[0.015/(0.0003)] | 0.97/(0.04)        | 1e6                       |
> | **Variant3**                                           | [0.01]/[0.01]                 | **[0.5]/[0.05]**            | [0.114/(0.01)]/[0.031/(0.001)] | 0.93/(0.05)        | 1e6                       |
>
> The results were obtained using three different random seeds and demonstrate that L2S consistently achieves optimal parameter values across the default setting and all tested variants.
>
> ***ManiSkill2-Open drawer***
>
> For this task, the parameter in the termination condition of the first skill specifies the required proximity of the robot's end effector to the target position above the drawer handle. In the variant, we increase the variance of this parameter from the default value of 0.02 to 0.05 to evaluate its impact on performance.
>
> | [Termination Func] | Initial Params Mean Value | Initial Params Variance | Optimized Params(std) | Success Rate/(Std) | Training Cost (Timesteps) |
> | ------------------ | ------------------------- | ----------------------- | --------------------- | ------------------ | ------------------------- |
> | Default            | [0.01]                    | [0.02]                  | [0.026(0.003)]        | 0.94/(0.06)        | 1e6                       |
> | **Variant**        | [0.01]                    | **[0.05]**              | [0.031(0.006)]        | 0.97/(0.02)        | 1e6                       |
>
> The results, obtained using three different random seeds, consistently demonstrate the robustness of L2S in handling various parameter variance configurations while maintaining effective performance.

---

> ### Author Response · Authors · 2024-11-27
>
> Thank you once again for your time and thoughtful review of our work. As the deadline of discussion phase approaches, please don’t hesitate to let us know if there are any additional questions we can address to further strengthen our submission.

---

> ### Comment · Reviewer_jBGA · 2024-11-28
>
> Thank you for addressing my questions. I have no further inquiries and am happy to stick with my original score!

---

### Official Review · Reviewer_pFie · 2024-11-04

**Soundness:** 2
**Presentation:** 3
**Contribution:** 2
**Rating:** 3
**Confidence:** 4

**Summary:**

This work introduces an LLM-guided skill discovery framework for solving robotic tasks. For each given task, LLM decomposes into sub-tasks, and writes reward and terminate functions for the RL agent to learn the corresponding policy.

**Strengths:**

The skills are designed with parameters, which have the flexibility to be reusable for other tasks.

**Weaknesses:**

- L2S depends heavily on the accuracy of LLM-generated reward and termination functions. Without evolutionary search or some kind of verification method, I doubt the effectiveness of this method on new tasks --- the current success relies on the expert few-shot prompt enumerating successful termination conditions.
- Since the aim of the method is to solve given tasks by decomposing them into sub-tasks, "skill discovery" may not be proper here.
- The experimental scenarios are simple. It would be valuable if diverse environments were shown.
- The approach has only been evaluated in state-based environments, limiting its applicability in vision-based or real-world robotic settings without additional adjustments.

**Questions:**

- How is the baseline being compared? Is the "Listing 3 Additional environment knowledge" also available to those baselines? If not, the comparison would be far from fair for Eureka.
- How to generalize to new environments, e.g. what effort do users have to put in?

---

> ### Author Response · Authors · 2024-11-25
>
> Thank you for your dedication to reading our work and providing valuable feedback. Below are our responses to the weaknesses and questions you raised.
>
> ## Questions
>
> > **How is the baseline being compared？**
>
> We compare our approach, L2S, against two baselines, Text2Reward and Eureka, using all available environment-related information. This includes an environment description, additional environment-specific knowledge, and few-shot examples. The "additional environment knowledge" component serves to bridge the semantics of real-world instructions with the simulated environment and clarify task success criteria. Specifically, this involves:
>
> 1. Mapping the 3D coordinate system of the environment to the human-provided task descriptions.
> 2. Defining the success conditions of tasks, which may vary based on human preferences.
>
> For example, in the LORL environment, a human instruction such as "move an object forward" is interpreted as:
>
> 1. Moving the object along the positive y-axis in the simulated environment.
> 2. Considering the task successful if the object is moved at least 0.1 meters along the positive y-axis.
>
> > **How to generalize to new environments？**
>
> To generalize to new environments, all environment-specific information in the prompt must be updated to align with the new environment, as the assets, supported tasks, and corresponding success conditions differ across environments. However, the "Instruction Hint" section remains consistent. For further details, please refer to the prompt blocks provided in Appendix E.
>
> ## Weaknesses
>
> > **L2S depends heavily on the accuracy of LLM-generated reward and termination functions. No evolutionary search**
>
> The primary contribution of L2S lies in its framework for leveraging LLMs to perform task decomposition and facilitate the efficient reuse of subtask policies across related task domains. In this work, we use few-shot examples to guide the LLM in generating accurate and stable free-form code for reward and termination functions. This design choice is inspired by related work, such as Text2Reward, which, however, does not address task decomposition or composition. Alternatively, we could replace the few-shot examples in our prompts with evolutionary search to guide the generation of reward and termination functions, as in Eureka. We believe that L2S would support skill discovery and reuse under Eureka's design strategy. However, our core contribution focuses on enabling skill discovery and policy reuse, rather than on the specific method—whether few-shot examples or evolutionary search—used for reward and termination function generation for subtask policies.
>
> > **The experimental scenarios are simple.**
>
> Our experiments prioritize tasks with compositional structures rather than the complexity of environment dynamics, as prior work (e.g., Text2Reward) has already demonstrated LLMs' capability to generate rewards in such environments. To this end, we compositionalized environments from Text2Reward, including robotics manipulation tasks in LORL and ManiSkill2, to highlight L2S's effectiveness in leveraging task compositionality for continuously presented tasks. For example, one of the long-horizon tasks in our experiments involves opening a drawer, placing a cube from the table into the drawer, and then closing the drawer. During the rebuttal phase, we further showcased an even more complex scenario where the agent must place three blocks into the drawer as part of the process (see our response to Reviewer 1Mn6). L2S not only solves these tasks significantly faster but also achieves higher success rates compared to Text2Reward.
>
> > **The approach has only been evaluated in state-based environments.**
>
> Our results demonstrate that L2S effectively decomposes complex tasks and generates accurate reward and termination functions for subtasks, which is the main focus of this paper. These reward and termination functions can principally be used by RL agents to train control policies using visual inputs and to terminate when specific visual conditions satisfy the termination criteria. Extending L2S to vision-based environments is left for future work.

---

> ### Author Response · Authors · 2024-11-27
>
> Dear Reviewer pFie, we hope this message finds you well. We hope the previous response would address the concerns and whether there is anything further we could clarify or elaborate on. Thank you.

---

### Meta-Review · Area_Chair_ENxe · 2024-12-17

**Metareview:**

Scientific Claims and Findings:
- The paper introduces L2S, a method that uses LLMs to decompose natural language task descriptions into reusable skills
- Each skill is defined by LLM-generated dense reward functions and termination conditions
- L2S builds a skill library progressively, using existing skills to reach states for learning new skills

Strengths:
+ Progressively building a skill library and using LLM to define their rewards are elegant ideas.
+ Strong empirical results across multiple challenging robotic manipulation tasks
+ Thorough ablation studies and evaluation methodology

Weaknesses:
- The core technical novelty is limited - the main contribution is replacing LEAGUE++'s pre-defined metric functions with LLM-generated rewards, while keeping the same fundamental approach of progressive skill learning and chaining.
- Heavy reliance on task-specific prompt engineering
- Lack of theoretical analysis for state abstraction method

Given ICLR's high standards for technical novelty, the incremental nature of the improvements over prior work (particularly LEAGUE/LEAGUE++) does not meet the bar. While the empirical results are strong and the engineering is solid, the paper lacks fundamental technical advances.

**Additional Comments On Reviewer Discussion:**

Initial Main Comments:
- Reviewer pFie raised serious concerns about the dependence on LLM-generated functions and limited experimental scenarios
- Reviewer jBGA questioned aspects of parameter optimization and variance selection
- Reviewer 1Mn6 highlighted significant overlap with prior work (League++) and concerns about LLM reliability
- Reviewer kCBF noted issues with prompt engineering requirements and limitations of the parameter optimization approach

The authors provided detailed responses addressing some concerns, including:
- Additional ablation studies on parameter distributions
- Clarification of the parameter optimization strategy
- New experiments on longer-horizon tasks
- Analysis of LLM function generation reliability

While these responses improved the technical clarity, they did not fully address the fundamental concerns about novelty relative to League++, the extensive prompt engineering requirements, and scalability limitations.

---

### Decision · Program_Chairs · 2025-01-22

Reject